# Autonomous Evaluation and Refinement of Digital Agents

**Jiayi Pan**[1]* **Yichi Zhang**[2] **Nicholas Tomlin**[1] **Yifei Zhou**[1] **Sergey Levine**[1] **Alane Suhr**[1]

[1]UC Berkeley  [2]University of Michigan

## Abstract

We show that domain-general automatic evaluators can significantly improve the performance of agents for web navigation and device control. We experiment with multiple evaluation models that trade off between inference cost, modularity of design, and accuracy. We validate the performance of these models in several popular benchmarks for digital agents, finding between 74.4 and 92.9% agreement with oracle evaluation metrics. Finally, we use these evaluators to improve the performance of existing agents via fine-tuning and inference-time guidance. Without any additional supervision, we improve state-of-the-art performance by 29% on the popular benchmark WebArena, and achieve around 75% relative improvement in device control settings. We release our code and data at https://github.com/Berkeley-NLP/Agent-Eval-Refine.

## 1 Introduction

Given an instruction, e.g., *"Tell me the cost of my latest canceled order,"* an automated digital agent would be expected to first navigate to a user's profile page, then to a list of their previous orders, identify the most recent order that has been canceled, and return its total amount to the user. Such agents offer the long-term potential of making digital devices more accessible, while also simplifying tedious or mundane tasks. However, in the short term, even state-of-the-art agents still make mistakes on simple tasks. Evaluating such agents and characterizing their failure modes is not only important for understanding and improving the models, but also critical for safely deploying them in real world. In this paper, we demonstrate the opportunities and efficacy of using automated evaluation models to both characterize and improve agent performance, without requiring access to any extra supervision, such as expert demonstrations or evaluation functions.

We propose to automatically evaluate user instructions and arbitrary agent trajectories with domain-general neural models. We explore two main variants of this approach (Figure 1, left): first, a modular caption-then-reason approach where a vision-language model (VLM) first captions the screenshots, and a language model (LM) is used to reason about if an agent succeeds based on textual information; and second, an end-to-end approach where we prompt an advanced VLM like GPT-4V (Achiam et al., 2023) to directly evaluate a trajectory. These two different approaches offer trade-offs in performance, cost, and transparency.

We first evaluate our proposed approach on its ability to match oracle evaluation metrics using WebArena (Zhou et al., 2024) and Android-in-the-Wild (AitW; Rawles et al., 2023), achieving accuracies up to 82.1 and 92.9% respectively. We then show how these evaluation models can be used to refine existing agents through inference-time guidance or during training, without access to any hand-designed evaluation functions or additional demonstration data (Figure 1, right). When integrated as the reward function in Reflexion (Shinn et al., 2023), the evaluator enhances the best-performing GPT-4 WebArena agent's success rate by up to 29% of relative improvement. Additionally, we evaluate in both Android and iOS device control settings. For iOS, there is no existing benchmark environment or training data. When using our evaluation models to filter sampled trajectories to be used in behavior cloning, we see relative improvements of around 75%.

---

*Email: jiayipan@berkeley.edu

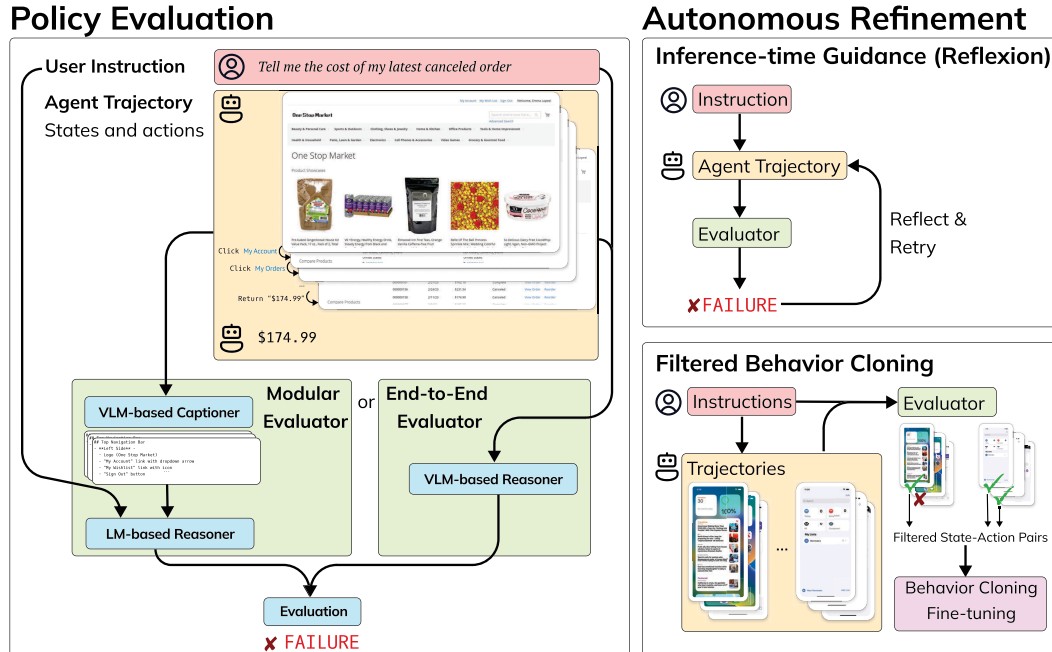

Figure 1: Method overview: A model-based evaluator provides evaluation of a digital agent's trajectory (left). It can be used as the reward function for Reflexion (Shinn et al., 2023) or filtered behavior cloning to enhance model performance (right).

## 2 Related Work

Building automated digital agents that map from user instructions to executable actions has been a long-standing goal in the NLP and AI communities (Allen et al., 2007; Branavan et al., 2009; 2010). Recent advances in NLP and multimodal machine learning have supported the development of more capable agents, and many recent benchmarks and approaches cover instruction-conditioned tasks such as web navigation and device control.

**Digital Agents**    Early modeling of language-conditioned autonomous agents focused on approaches that include semantic parsing (Allen et al., 2007; Xu et al., 2021; Li et al., 2020), reinforcement learning (Branavan et al., 2009; 2010), and imitation learning (Humphreys et al., 2022). The strength of pretrained language and language-and-vision modeling has renewed interest in building language-conditioned digital agents (Zhang & Zhang, 2023; **?**; Zhou et al., 2024; Deng et al., 2023; Wang et al., 2023a; Gur et al., 2024). For example, baseline approaches to WebArena (Zhou et al., 2024) use few-shot prompting with language-only models, representing the environment state and action space with its document object model (DOM). More recent works in building these agents have moved from language-only modeling to vision-language modeling, representing the environment state space as its rendered pixel representation instead of relying on a DOM. Another line of work has applied inference-time techniques to improve model's performance, for example with inference-time exploration (Zhang et al., 2023), intermediate plan revision (Zhang et al., 2024) and error correction (Wang et al., 2024), and self-critique (Wu et al., 2024) on GPT-4 or GPT-4V. Concurrent to our work, OS-Copilot (Wu et al., 2024) proposes a self-critique component to autonomously refine Mac device control agents, implementing the critic as a LM that reasons about proposed tool implementations and error messages. In contrast to our work, this critic does not evaluate actual agent behavior in the execution environment or used in model training.

**Autonomous Refinement and Evaluation**    Recently, there has been renewed interest in methods for improving policies at training (Ouyang et al., 2022; Bai et al., 2022; Lee et al., 2023; Abdulhai et al., 2023) or inference (Shinn et al., 2023; Yao et al., 2023; Wu et al., 2024)

time without human supervision. Approaches like Reflexion (Shinn et al., 2023) assume access to an external evaluation function, leveraging it as the supervision signal to guide policy improvement at inference time. In contrast, we study applications of inference-time autonomous refinement without requiring access to the external evaluation function, and show that using our proposed domain-general evaluation models improves agent success rate by 29%. Meanwhile, methods which bootstrap policies with supervised training and refine them with reinforcement learning have been widely adopted; our proposed evaluators enable this paradigm in open, realistic digital agent scenarios, providing a relative improvement in performance of over 70%. Concurrent to our work, WebVoyager (He et al., 2024) also explores using GPT-4V as an automated proxy for human evaluation of web agents, though is not the primary focus of their work, and neither performs in-depth analysis of the quality of its judgments, nor explores its applicability to improving agents. More broadly, model-based metrics have been highly successful across various domains, such as evaluating image or text quality (Heusel et al., 2017; Zheng et al., 2024), and assessing success of a robot or human activity (Du et al., 2023). Part of our contribution is to provide strong empirical evidence to extend this success to the domain of real-world digital agents.

**Digital Agent Benchmarks** Recently-proposed benchmarks that study digital agents fall roughly into two categories: simulation-based benchmark and demonstration-based one. Simulation-based benchmarks include environment simulators that offer the ability to execute arbitrary agent trajectories. Early simulation environments such as WoB (Shi et al., 2017; Liu et al., 2018), WebShop (Yao et al., 2022), and others (Branavan et al., 2009) are limited in their domain coverage, realism, or generalizability of their evaluation functions. Recently proposed simulation environments like AndroidEnv (Toyama et al., 2021), WebArena (Zhou et al., 2024) and VisualWebArena (Koh et al., 2024), though far from perfect, have offered improvement across these dimensions. However, designing simulators, curating tasks, and handcrafting evaluation functions fundamentally limits their ability to mirror task and environment diversity of real environments.

In parallel, the community has focused on demonstration-based benchmarks that do not include an executable simulation environment, including PIXELHELP (Li et al., 2020), MoTIF (Burns et al., 2022), Mind2Web (Deng et al., 2023), and AitW (Rawles et al., 2023). Notably, Mind2Web and AitW contain over 2K and 715K human trajectories respectively on a wide range of web navigation and device control tasks. Though primarily used for model training (Rawles et al., 2023; Deng et al., 2023; **?**; Zhang & Zhang, 2023), these datasets are also used for evaluating digital agents through reference-based metrics like action matching score. In this setting, an agent is given the prefix of a human demonstration and evaluated on its prediction of the next action to take. However, this metric requires human demonstrations and does not directly reflect agent's performance in real-world because it does not account for consequences of an agent's sequential decision process, alternative actions that diverge from the demonstration.

We propose a third approach in which arbitrary instructions and agent trajectories are directly evaluated by a model.

## 3 Domain-General Evaluators

We develop multiple domain-general automatic evaluators for digital agents. Given a user instruction $x$ and an initial environment state $s_0$, an agent generates and executes a sequence of actions $\bar{a} = \langle a_0, a_1, \ldots, a_n \rangle$, resulting in a sequence of state visits $\bar{s} = \langle s_0, s_1, s_2, \ldots, s_{n+1} \rangle$. In this work, we assume $a$ and $x$ are in text form, such as `<Type:''Hello''>` and *"Check the weather"*, and each state $s$ is represented as a screenshot image. Given $x$, $\bar{a}$, and $\bar{s}$ as input, the model produces a scalar evaluation $\bar{r} = \langle r_0, r_1, \ldots, r_n \rangle$ corresponding to each step of the trajectory:

$$\bar{r} = \text{evaluate}(x, \bar{a}, \bar{s}) .$$

The evaluator can provide either trajectory-level or per-step evaluations. For trajectory-level evaluation, $r_0 = \cdots = r_{n-1} = 0$, with $r_n = 1$ for successful trajectories and $r_n = 0$ otherwise. For per-step evaluation, we classify each step into three types, $r_i = 1$ indicates task success after action $a_i$, $r_i = p \geq 0$ indicates progress toward the goal, and $r_i = d < 0$ is assigned to

actions that do not contribute to the objective. We query the model once for trajectory-level evaluation and $n$ times for per-step evaluation, reducing the model's task into a binary or ternary classification problem at each step.

We explore two methods for constructing the model:

1. An end-to-end approach that maps directly from instructions and screenshots to an evaluation via a pre-trained VLM.

2. A modular approach which first transcribes the observed screenshots into text descriptions using a VLM, and then uses a LM to map the descriptions, actions, and user instruction onto an evaluation.

Both methods have tradeoffs: in the first, we can apply advanced VLMs like GPT-4V. However, this approach is relatively expensive and relies on API calls to proprietary models. In the second, we can compose open-weight models to achieve slightly weaker performance, but with added benefits of explainability via modularity and low-cost local deployment.

### 3.1 End-to-End Approach

We directly provide an instruction-tuned VLM with $x$, $\bar{a}$, and $\bar{s}$. We prompt it to first produce a text-based reasoning process (Wei et al., 2022), then output its evaluation result. In our experiments, we use the proprietary vision-language model GPT-4V (Achiam et al., 2023).[1]

### 3.2 Modular Caption-then-Reason Approach

Many existing approaches for joint reasoning about language and vision disentangle perception and reasoning. In these approaches, a VLM is first applied to visual input to generate a language-based description; then, a text-only model (e.g., a LM) takes as input this description and the user instruction to produce a response by reasoning only about linguistic inputs. Existing work applying this approach has mostly focused on joint reasoning about natural images and text, e.g., for visual question answering (Guo et al., 2023; You et al., 2023; Wang et al., 2023b). We take a similar approach here, where we first use a VLM to produce a description of the agent's observations given as $\bar{s}$, then feed these descriptions, along with actions $\bar{a}$ and the user's instruction $x$ to an LM to produce a final evaluation.[2]

**Captioner** One drawback to this modular approach is the potential for information loss, where the image description may not include all the details necessary for task success (Wang et al., 2023b). In our case, this could include missing or misrepresenting details about the screenshot, and indeed, we find that current open-weight VLMs struggle to produce detailed screenshot descriptions out of the box. In contrast, the most advanced, yet proprietary, VLMs can produce very detailed descriptions with adequate prompting.

To improve a captioner's ability to provide detailed, well-formatted descriptions, we collect a dataset of screenshots paired with descriptions, and use it to fine-tune an open-weight VLM. We first acquire screenshots from a variety web and device control domains, then use GPT-4V to provide an initial detailed description for each screenshot. We manually filter out or fix apparent errors in GPT-4V's output, resulting a total of 1,263 data points.[3] We use this data to fine-tune the QWen-VL-chat (Bai et al., 2023) model. During both finetuning and at inference time, we provide text recognition results from EasyOCR[4] as an additional input to the VLM to reduce hallucination.

At inference time, we use our finetuned captioner model to acquire a description for each step in the agent trajectory. Critically, we do not provide this model access to the original user instruction, as we find this exacerbates model hallucinations; e.g., describing webpage attributes which would be relevant to the task, but are not actually present in the screenshot.

**Reasoner** Finally, we provide the actions, generated descriptions, and the original user instruction to a language-only instruction-tuned model. We experiment with prompting two

---

[1] Prompt templates and additional details are provided in Appendix A.1.

[2] Data collection process, hyper-parameters, and output examples are detailed in Appendix A.2.

[3] Table 3 in Appendix A.2 contains details of data sources and sizes.

[4] https://github.com/JaidedAI/EasyOCR

LMs, Mixtral (Jiang et al., 2024) and GPT-4, to produce a text-based thought and reasoning process as well as the final evaluation.

## 4 Experiments and Results

Our goal is to show how domain-general evaluation models can support the autonomous evaluation and refinement of digital agents, without requiring access to human demonstrations or oracle evaluation metrics. To this end, we first evaluate how these models perform **as autonomous evaluators** by comparing their judgments to benchmark-provided metrics and human judgements (Section 4.1). We then illustrate how these evaluation models, while imperfect, can serve **as discriminators in autonomous refinement settings** through both inference-time policy refinement (Shinn et al., 2023) and filtered behavior cloning (filtered BC; Chen et al., 2020; 2021; Emmons et al., 2022) to support significant improvements in agent performance (Section 4.2).

Our rationale behind experiment design is to cover a broad range of domains and challenges. We use WebArena for both evaluation and inference-time refinement, as its built-in evaluation functions facilitate direct comparison. Android-in-the-Wild (AitW) is chosen for evaluation since it is widely used for training and evaluating Android agents, and is typically evaluated using a reference-based metric instead of task success. Next, we refine a model through filtered behavior cloning on iOS, where data scarcity poses a significant challenge to supervised methods. We further validate the effectiveness of filtered behavior cloning with a similar, larger experiment on Android.

**Environments** WebArena (Zhou et al., 2024) is an offline web emulation environment and dataset that supports execution of arbitrary policies. WebArena comprises 812 human-written task instructions across various domains, including shopping, maps, and content management systems. Each instruction is paired with a handwritten test case that verifies agent success, e.g., by checking the status of a specific webpage element against a reference. We refer to this set of test cases as WebArena's oracle evaluator.

Android-in-the-Wild (AitW; Rawles et al., 2023) is a large-scale dataset for Android device control containing 715,142 human demonstrations of 30,378 unique instructions. In our experiments, we focus on a subset of 120 tasks randomly sampled from the AitW test set.[5] Unlike WebArena, AitW does not include an emulation environment for agent execution. Instead, the suggested evaluation metric is based on action matching: given a sequence of actions representing the prefix of a human demonstration, the agent is evaluated on its ability to predict the next action in the demonstration. While we compare against this reference-based metric in our experiments, we focus on end-to-end task-level success rate and implement an Android emulator to support execution of arbitrary trajectories.[6] We refer to human judgements on trajectory success as the oracle evaluation.

Despite significant interest in developing digital agents, progress in the domain of iOS device control has been modest, with the exception of Yan et al. (2023), who collect a small unreleased dataset of human demonstrations in this domain. We curate a set of 132 tasks in the iOS domain, taking inspiration from tasks included in AitW. We experiment with using our proposed evaluation models to facilitate domain transfer, with the goal of applying the strongest model on AitW, CogAgent (**?**), to iOS. We develop a Python interface to the iOS emulator on macOS, and design its action space to align with the Android-in-the-Wild to facilitate domain transfer.[6]

**Evaluation Models** We evaluate three evaluation model variants:

- GPT-4V: End-to-end approach (Section 3.1) using GPT-4V.
- QWen-VL-chat: End-to-end approach (Section 3.1) using QWen-VL-chat. This serves as open-weight model's baseline performance.

---

[5]We subsample from the original test set of 1.4k tasks to facilitate acquiring human judgments of trajectories. See Appendix A.4 for details on a list of evaluated tasks and details on task sampling.

[6]Details on our emulators are available in Appendices A.4 and A.6.

|  | GPT-4V | QWen-VL-chat | Captioner + Mixtral | Captioner + GPT-4 |
|---|---|---|---|---|
| WebArena (%) | 80.6 | 68.0 | 74.4 | 82.1 |
| Android (%) | 90.6 | 70.2 | 92.9 | 89.8 |

Table 1: Comparison of evaluators accuracy against oracle evaluator or human judge in WebArena and Android.

- Captioner + Mixtral: Modular approach (Section 3.2) using a finetuned QWen-VL-chat (Bai et al., 2023) to generate a trajectory description, and Mixtral (Jiang et al., 2024) to provide the final evaluation.
- Captioner + GPT-4: Modular approach (Section 3.2) using a finetuned QWen-VL-chat to generate a trajectory description, and GPT-4 to provide the final evaluation.

In most experiments, the evaluation model produces a trajectory-level evaluation, and takes as input only the last frame $s_{n+1}$ in the trajectory, along with the instruction $x$ and action sequence $\bar{a}$. Preliminary experiments suggested that model performance does not improve with information about previous states, likely due to limitations of existing models in processing long contexts. In the iOS experiments, the evaluation model takes as input the entire trajectory $\bar{s}$ and $\bar{a}$ and the instruction $x$, and produces a per-step evaluation.

**Agent Policies**   We experiment with evaluating and refining the current state-of-the-art digital agents. In WebArena, this is a GPT-4-based agent described by Zhou et al. (2024). For each task, GPT-4 is provided the user's instruction and the current DOM representation of the webpage derived from its HTML accessibility tree. GPT-4 is prompted to generate an action grounded in the DOM, e.g., clicking a button with a specific element ID. The task success rate of this agent is 14.4% as reported in Zhou et al. (2024), whereas our reproduction, as shown in Figure 3, indicates a success rate of 15.6%.

The strongest agent on the AitW benchmark is CogAgent (**?**), followed by Auto-UI$_{\{large, base\}}$ (Zhang & Zhang, 2023). These agents are implemented as neural vision-language models that map observations, represented as images, and instructions to executable actions. We also experiment with the human demonstrations provided in AitW.[7]

### 4.1   Automatic Evaluation

**WebArena**   For each WebArena task and corresponding trajectory sampled from the GPT-4-based policy (Zhou et al., 2024), we acquire task-completion judgments for each of the three evaluation systems described above. Table 1 shows the overall accuracy of the evaluator's predictions.[8] The end-to-end approach with GPT-4V achieves 80.6% accuracy, while Captioner + Mixtral, which uses only open-weight models, matches the oracle's evaluations for 74.4% of tasks, and replacing Mixtral with GPT-4 achieves the highest accuracy at 82.1%. Additionally, we see that using QWen-VL-chat end-to-end significantly underperforms the modular, open-weight Captioner + Mixtral approach, confirming the effectiveness of our modular design choice.

**Android-in-the-Wild**   For the 120 sampled test tasks in AitW, we evaluate trajectories sampled from four policies: CogAgent (**?**), Auto-UI$_{\{large, base\}}$ (Zhang & Zhang, 2023), and human experts (Rawles et al., 2023).[7] We acquire human judgments of trajectory success, as well as judgments from the three evaluation model variants.[9]   Figure 2 shows the performance of all four agents as evaluated by humans and the three evaluator variants.

---

[7]The human demonstrations use the original AitW emulator, which was not released by the authors; thus, these results are not directly comparable with the automated policies, which use the emulator we implement. However, the focus of our experiments is not to directly compare policies, but to compare evaluators across a variety of policies, tasks, and domains.

[8]Figure 4 in Appendix A.3 includes the confusion matrices of these predictions.

[9]Experimental setup details for AitW are provided in Appendix A.4.

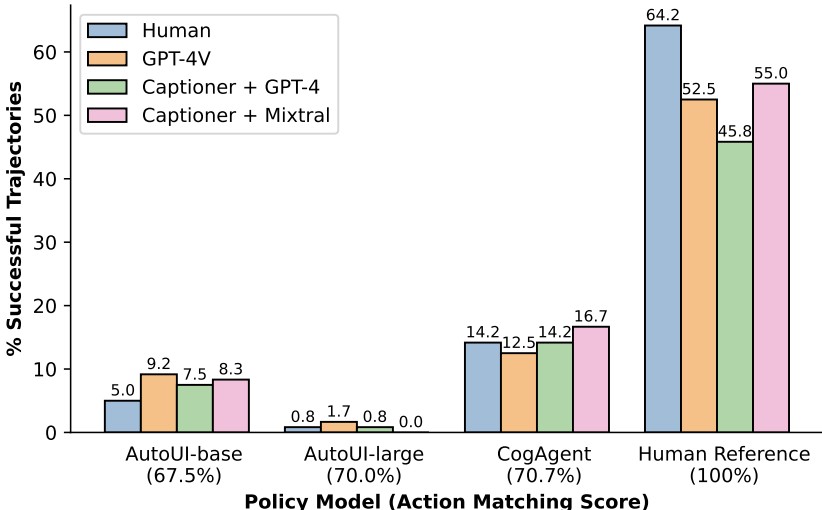

Figure 2: Evaluating models in Android-in-the-Wild with different evaluation methods. We use human judgments of trajectory success as oracle reference and compare it with judgments from our evaluation models and AitW's standard action matching score.

Below each agent label we also include each policy's partial action match score (Li et al., 2020), which is the standard reported metric for agents on AitW.[10]

Unsurprisingly, we find that the human reference trajectories achieve the highest performance as evaluated by all success metrics. However, our analysis reveals that about 36% of the human demonstrations we annotate are actually unsuccessful, with common errors including early stopping, completing the wrong task, and making mistakes with respect to the parameters of the task. The difficulty of collecting high-quality demonstration data at scale further demands automated evaluation methods that can either act as a quality filter or provide more direct evaluation than action matching score.

Among the three neural network policies, CogAgent achieves the highest success rates, followed by Auto-UI$_{base}$, while the performance of Auto-UI$_{large}$ is close to zero according to all evaluators. When comparing conclusions that can be drawn from the two styles of metrics – task success and action matching – there are three clear differences: first, that success rate lags far behind single-step action prediction; second, that relative performance of models changes depending on the metric used; and third, that using a reference-based metric on erroneous references could result in inflated impressions of model performance. In particular, while Auto-UI$_{large}$ appears to outperform Auto-UI$_{base}$ according to the action matching metric, it is clearly inferior in terms of overall task success rate. Quantitatively, all three evaluators achieve a Kendall correlation of 100% with the human judges, while the action matching score only obtains 66.7%. This highlights a fundamental drawback in a single-step metric like action matching: it does not reflect error propagation or distribution shift in the sequential prediction process of an arbitrary policy, which can be captured by whole-trajectory success metrics.

Measuring whole-trajectory success for the complex tasks that digital agents complete has typically required either human evaluation of individual trajectories, or manual creation of individual test cases, as in WebArena. We analyze the potential for automating this process using our three proposed evaluators. Table 1 shows the accuracy of each evaluator variant aggregated over trajectories from all four policies.[8] Overall, we find that our automated metrics correlate very strongly with human judgment: the Captioner + Mixtral variant shows the highest agreement with human judgment at 92.9% accuracy; replacing Mixtral with GPT-4 leads to a performance drop to 89.8%; and the end-to-end approach of GPT-4V

---

[10]Action matching scores are averaged across the subsets of AitW we sample from, as reported in **?** and Zhang & Zhang (2023).

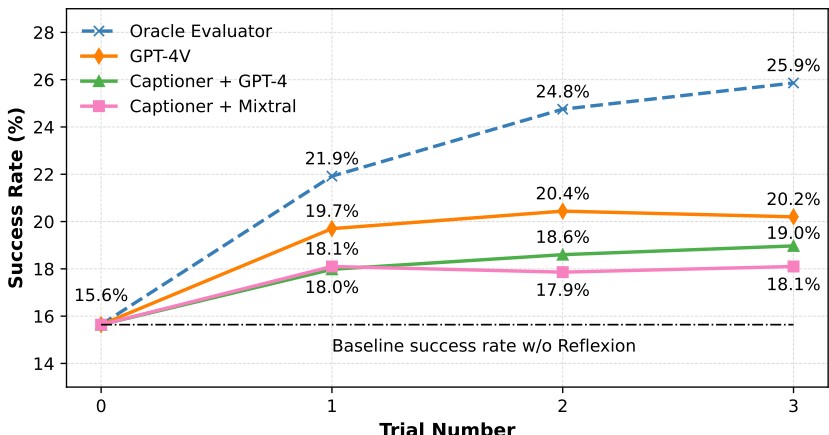

Figure 3: Results of applying Reflexion for up to 3 rounds using different evaluation systems on the WebArena benchmarks. Here, the oracle evaluator denotes performance using WebArena's built-in evaluation functions as the reward function; this provides an upper-bound of improvement using Reflexion.

achieves 90.6% accuracy. Again, the low accuracy with QWen-VL-chat confirms our modular approach design choice.

## 4.2 Autonomous Refinement

**Reflexion on WebArena** We demonstrate how our proposed evaluation models can serve as a reward signal to guide an existing web agent at inference time, using the Reflexion technique (Shinn et al., 2023) as an example. In Reflexion, an agent first attempts a task, and an external evaluator is used to judge whether its attempt was successful or not. If it is judged as unsuccessful, the agent will be prompted to reflect on the failure and retry. We experiment with improving the current state-of-the-art GPT-4-based WebArena agent.[11]

Figure 3 includes the agent's baseline performance, and performance using up to three rounds of Reflexion with the oracle evaluator (which serves as an upper bound) and our three evaluation systems as external supervision. We see the improvement our evaluators provide scales favorable with evaluator capability, with Captioner + Mixtral improves agent's relative success rate by 16% and GPT-4V based evaluator by 29%. All system variants, including the low-cost and locally-hosted variant Captioner + Mixtral, significantly enhance agent's performance while requiring no access to hand-designed evaluation functions.

Our preliminary study suggests that false negative evaluations have a more detrimental impact on agent's performance compared to false positives. If our evaluator predicts an execution is incorrect, but it was actually successful, this forces the agent to retry a successful execution, which nearly always leads a subsequent failure. In contrast, false positives only lose out on the opportunity to retry, which creates an upper bound of performance for the agent, but does not degrade its performance. Improving the robustness of inference-time algorithms under noisy supervision is an interesting future direction to explore.

**Filtered Behavior Cloning for Device Control** We demonstrate how our evaluator can guide the refinement of a policy using filtered behavior cloning (filtered BC), without additional supervision. We experiment on both iOS and Android environments.

On iOS, we use CogAgent (**?**) as the policy model for the experiment. It is primarily instruction-tuned with demonstrations from web navigation and Android device control, and incorporates a very limited, manually collected iOS dataset for training. For data collection and testing purposes, we design 132 common tasks on iOS, with 80 tasks for training and 52 for testing. Given scaling limitations of emulation, including low speeds

---
[11]Reflexion prompts are detailed in Appendix A.5.

and restriction to emulation on macOS, we only experiment with iOS built-in apps and with the Captioner + Mixtral evaluator.

We first sample 737 trajectories from CogAgent, conditioned on the 80 training tasks. We use our evaluator to provide per-step evaluations to these trajectories, then apply filtered BC for fine-tuning using this data. Unlike standard fine-tuning, this method filters out data points with rewards below a specified threshold. We set this threshold at $\geq p$; i.e., we retain only state-action pairs that positively influence the success of a trajectory (Section 3). Additionally, we assess CogAgent's unmodified performance on iOS and explore a self-training approach by finetuning without data filtering as baselines for comparison.

Table 2 contains results for the 52 test tasks. iOS device control is a challenging task, with the baseline agent completing only 8 out of 52 tasks, yielding a 15% success rate. Self-training improves over the baseline by 3 tasks. Filtered BC with our evaluator significantly improves the policy model's performance from 8 to 14 successes, marking a 75% relative improvement. We also conduct a preliminary analysis of our per-step evaluator's accuracy. Human annotators agree with our step-wise evaluator on 43 of 50 state-action pairs randomly sampled from the iOS experiments.

However, our iOS experiments require physical Apple computers for emulation, which are slow and not scalable, leading to natural limitations in scope of our evaluation. To address this, we design a similar but larger experiment on Android. Unlike iOS, Android allows us to use scalable emulators on clusters, enabling efficient repetition of experiments. We use AutoUI-base policy for the experiment, selecting 560 tasks for training and 96 for evaluation. The base policy (AutoUI-base) completes 15 tasks, improving to $18.9 \pm 1.0$ (averaged over 3 training runs) with self-training. Filtered BC with our step-wise evaluator still significantly improves, succeeding at $26 \pm 0.8$ tasks,r esulting in a 73% relative improvement.

| Policy | # Success (Total: 52) |
|---|---|
| CogAgent | 8 |
| + Self-training | 11 |
| + Filtered BC (Ours) | 14 |

(a) iOS Experiment

| Policy | # Success (Total: 96) |
|---|---|
| Auto-UI-base | 15 |
| + Self-training | $18.9 \pm 1.0$ |
| + Filtered BC (Ours) | $26.0 \pm 0.8$ |

(b) Android Experiment

Table 2: Comparison of base policy and refined policies via self-training and filtered behavior cloning, including the number of successful tasks in our test set.

### 4.3  Error Analysis

We randomly sample 20 successful and 30 erroneous evaluations for each evaluation model in WebArena and manually annotate the sources of failure.[12] We categorize errors into three primary types, providing percentage estimates rounded to the nearest 5%.

1. Critical information lost from captions in the modular approach (10%); errors in screenshot understanding for the end-to-end GPT-4V approach (5%).

2. Errors in the reasoning process, observed in 50% of cases for GPT-4V/GPT-4-based methods and 70% for Mixtral-Captioner.

3. Ambiguities in task specification and success criteria, observed in 30% of cases for GPT-4V/GPT-4-based methods and 10% for Mixtral-Captioner.

We note that in our error categorization, a model must overcome errors in preceding categories to be assessed under the subsequent one. Consequently, Mixtral-Captioner's lower rate of Type 3 errors is mostly attributed to its higher frequency of Type 1 and 2 errors.

Additionally, we find the model provides the correct final evaluation, but incorrect reasoning, in about 10% of correct evaluations.

---

[12]Refer to Figures 5 through 11 in the Appendix for visual representations of these evaluations.

## 5 Conclusion

In this study, we design automatic methods to both evaluate and refine the performance of digital agents. We first describe a model that provides either trajectory-level or per-step evaluation of agent's performance. Subsequently, we propose two approaches to implement the model: an end-to-end approach using a pre-trained vision-language model, and a modular caption-then-reason approach using a VLM and a pre-trained language model together. These methods offer trade-offs between performance, cost, and modularity.

Using WebArena and Android-in-the-Wild as testbeds, we first validate the effectiveness of these evaluators against oracle evaluation metrics, and highlight their advantage over standard reference-based metrics on AitW. We then show how the evaluators can be used to refine existing agents through both inference-time guidance and filtered BC. When integrated as the reward function in Reflexion, a method for inference-time refinement, our evaluators enhance the best-performing agent's success rate by up to 29%. Additionally, it boosts the performance of a strong device control policy in a domain transfer task by 75% via filtered behavior cloning, all without any extra supervision. Our findings show the potential of model-based automated evaluators for both evaluating and improving digital agents, which is especially critical in developing real-world agents where ground truth evaluation functions or human supervision are not always available.

## Limitations and Future Work

While our research demonstrates the potential of model-based evaluators in evaluating and improving digital agents, we also identify several areas for future exploration. First, current evaluators are still far from perfect, and any enhancement in their performance, e..g, from better representations of the action space or stronger base models, will likely directly translate to improved outcomes. Second, in this work, we focused on Reflexion and filtered behavior cloning. Future works can explore scaling up the experiments and developing better training and inference-time algorithms that are robust and efficient under noisy supervision. Finally, in this work we only make use of the evaluator's binary or ternary judgment, and discard the language-based explanation it generates. Future work can explore how to leverage this information, for example, to further enhance policies through language supervision or to provide scalable oversight of agent behavior.

## Ethics Statement

Most currently available digital agents are research artifacts. As the performance of these agents improve and they are increasingly deployed in the real world, they may pose security risks to their users. For example, a web agent with unconstrained access to a browser might be able to gain access a user's passwords, financial information, or social media messages. Better understanding the potential failure modes of these models in real-world use cases is critical to ensuring their safe deployment. We view our work as a first step in this direction: by developing domain-general evaluators, we hope to facilitate better understanding of models (and their risks) outside of simulated environments like WebArena. At the same time, human evaluation and oversight of these future systems will also be important for mitigating potential harms; although our work in this paper focuses on autonomous evaluation, we hope it will supplement, rather than supplant, human efforts.

### Acknowledgments

We thank the Berkeley NLP group, especially Ruiqi Zhong, Andre He, Charlie Snell, Catherine Chen, Sanjay Subramanian, and Zineng Tang, as well as Allen Nie for feedback and discussions and Shuyan Zhou for assistance in setting up the WebArena experiments. This work was partially supported by an AI2 Young Investigator Grant. NT is supported by the DARPA SemaFor program.

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

# A    Experiment Details

In this section, we provide details about our experiments. Please refer to our code at `https://github.com/Berkeley-NLP/Agent-Eval-Refine` for the official reference.

## A.1    End-to-End Approach

We use `gpt-4-1106-vision-preview` through the OpenAI API and feed the image without resizing in "high-resolution". We use a temperature of 0 and keep other parameters at default. The prompt templates for each environment are provided in Figures 12 and 13.

## A.2    Modular Caption-then-Reason Approach

**Collecting screenshots**    As described in Table 3, we constructed our dataset primarily through random subsampling from source datasets. However, for the iOS domain, due to limited online resources, we manually capture 50 extra screenshots in-house.

**Action representation**    We represent actions as strings, e.g., `Type ''Hello''` . This method leads to information loss when processing actions like clicks for pixel-based policies, as the coordinates `[x, y]` become meaningless when the image is represented by its textual description. We leave the task of more adequately transforming pixel-localized actions into textual forms for future work.

**Collecting screenshot descriptions**    After obtaining the screenshots, we query GPT-4V (specifically, `gpt-4-1106-vision-preview` through the API) to get dense caption demonstrations. We manually fix or filter out ones with apparent errors. We use a temperature of 0 and keep the other parameters at their default settings. The prompt template is provided in Figure 15.

**Finetuning Qwen-VL captioner**    The prompt template to query the finetuned Qwen-VL-chat captioner is provided in Figure 16. We fine-tuned the model over 3 epochs with a batch size of 72 and adamw optimizer (Kingma & Ba, 2014), employing a cosine scheduler for learning rate adjustments starting from 1e-5, a weight decay of 0.1, and a warmup ratio of 0.01. As shown in the prompt template, during both finetuning and at inference time, we provide text recognition results from the EasyOCR engine as an additional input to the model to reduce hallucination. We provide randomly-sampled model output examples in Figure 14.

**Quering the reasoner**    After obtaining the descriptions of the screenshots and actions, we query the LM, either `Mixtral-8x7B-Instruct-v0.1` or `gpt-4-turbo-preview`. We use a temperature of 0 and keep the other parameters at their default settings. We provide the prompts to query the trajectory-level evaluator on Web and Android, and step-wise evaluator on iOS in Figures 17, 18, and 19 respectively.

## A.3    Evaluation on WebArena

We directly use the `GPT4-0613 + CoT - v2` trajectories released by WebArena for evaluation. Confusion matrices for our evaluators' predictions compared to the oracle evaluator are shown in Figure 4 (left).

## A.4    Evaluation on Android

**Emulator**    We use Android Studio's built-in emulator to simulate a Pixel 4 with API version 33, and we develop a Python API for agent execution based on the appium package. We opt not to use AndroidEnv for Android emulation as it lacks support for typing actions.

**Tasks**   The 120 evaluation tasks are evenly and randomly sampled from the General, WebShopping, and GoogleApps subsets of the Android-in-the-Wild test set (40 each) as shown in Listing 1. Note that we have excluded the Install and Single subsets. The install tasks require credit card information and are not safe to evaluate, while single-step tasks fall outside our focus on trajectory-level tasks.

**Evaluation**   We use temperature of 0, for all policies during evaluation. The confusion matrices comparing our evaluators with human judgments are presented in Figure 4 (right).

## A.5   Refinement on WebArena

**Reflexion**   We implement the Reflexion agent following the original paper (Shinn et al., 2023). The algorithm involves three key components: an Actor, an Evaluator, and a Self-Reflection module. The Actor generates thoughts and actions in text based on the state observations it receives, where the actions are parsed into executable commands to transform the environment. The Evaluator assesses the quality of the outcome produced by the Actor. It computes a reward score based on how well the generated trajectory align with the goal. If the evaluator assesses the task to be failed, the Self-Reflection model will be evoked to generate verbal reflections, which is stored in the actor's memory, facilitating improved decision-making in subsequent trials.

**Impelementation Details**   We use the DOM tree representation from the WebArena simulator as the environment observation. The LLM we use for Actor and the Self-Reflection is the `GPT-4-preview-1106` model, and the prompts for these are shown in Listing 3 and 4 respectively. For the evaluator, we experiment with all the three variants proposed as well as the oracle evaluator from WebArena which is used for performance evaluation. Note that we use the webpage snapshot images instead of the DOM tree as the input to our evaluator.

## A.6   Refinement on iOS

**Emulator**   We use XCode's built-in emulator to emulate an iPhone 13 device running iOS 16 and develop a Python API for the agent based on the facebook/idb package. We align its action space with that of the Android-in-the-Wild schema. Notably, since swiping up on the homescreen in Android means opening the AppDrawer, which can roughly be translated to swiping left or right on the homescreen, we bridge this domain gap by translating swiping up to swiping left or right 50% of the time during data collection and to the right 100% of the time during evaluation.

**Tasks**   As shown in Listing 2, we design 132 task instructions that covers typical iOS device control tasks on Apple's official apps, where 80 are used for data collection and training the agent, and 52 are hold out for testing. While we aimed to minimize the distribution difference between our tasks and that from the AitW dataset, the inherent differences between the platforms necessitated the inclusion of iOS-specific instructions, such as "Disable Siri's access to Photos in Settings."

**Filtered BC Details**   We finetune the CogAgent model with its official code and apply LoRA (Hu et al., 2022), setting the parameters as follows: lora rank to 50, employing a cosine learning rate schedule, with a warmup proportion of 0.2, a learning rate ($lr$) of 0.00001, a batch size of 4 and train for 3000 steps. We use a releatively high temparature of 1.5 and topk=100 during data collection to improve diversity and greedy decoding (temperature=0) for all policies during evaluation.

## A.7   Refinement on Android

We reuse the Android emulator as described in Appendix A.4. and sample the training and evaluation tasks from AitW General split's corresponding part. We use AutoUI-base as the policy. We set the temperate for 1 during sampling and uses a learning rate of $3e-3$ and a batch size of 128 for the training.

| Source | # | Domain |
|---|---|---|
| WebScreenshot (Dwyer, 2020) | 128 | Web |
| Mind2Web (Deng et al., 2023) | 429 | Web |
| AitW (train set) (Rawles et al., 2023) | 596 | Android |
| GPT-4V in Wonderland (Yan et al., 2023) | 60 | iOS |
| In-house | 50 | iOS |
| Total | 1,263 | |

Table 3: Sources of the screenshots dataset, including the number of screenshots paired with detailed descriptions and the original domain of the screenshots.

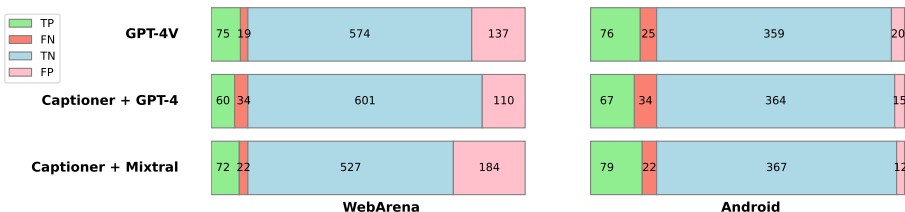

Figure 4: Confusion matrices for different evaluation systems against oracle evaluator or human judge on WebArena and Android.

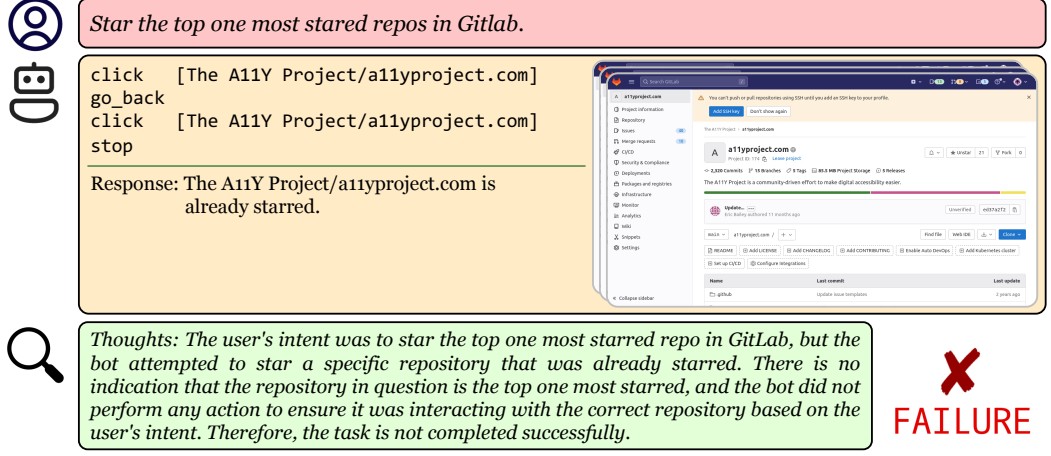

Figure 5: The evaluator detects a hard-to-find issue caused by a key missing step in the execution (i.e. ordering).

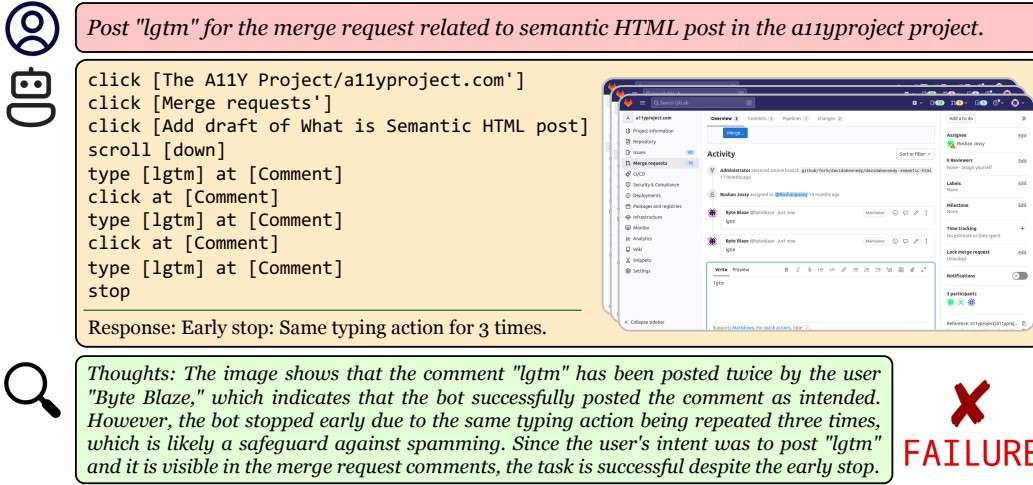

Figure 6: The evaluator correctly judge a task execution as successful despite the task ends with an exception.

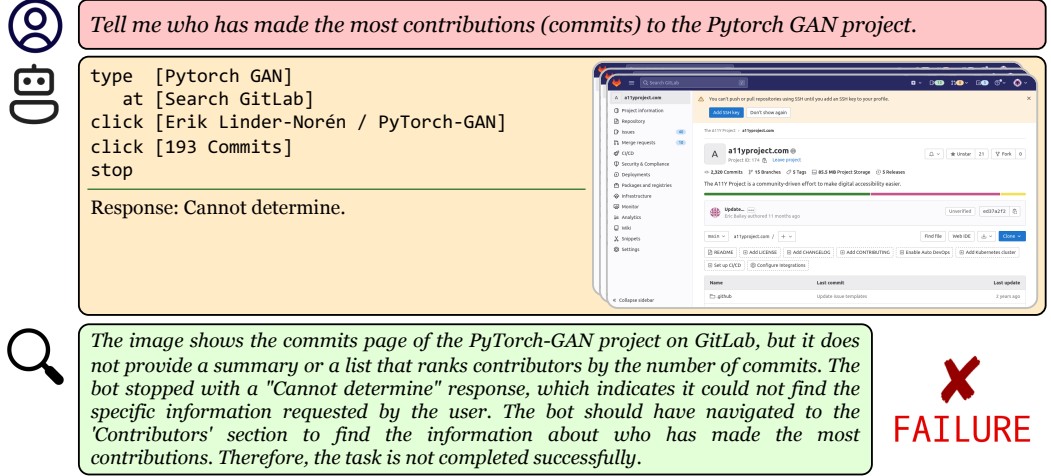

Figure 7: The evaluator can also provide insight for possible corrective actions of failure executions.

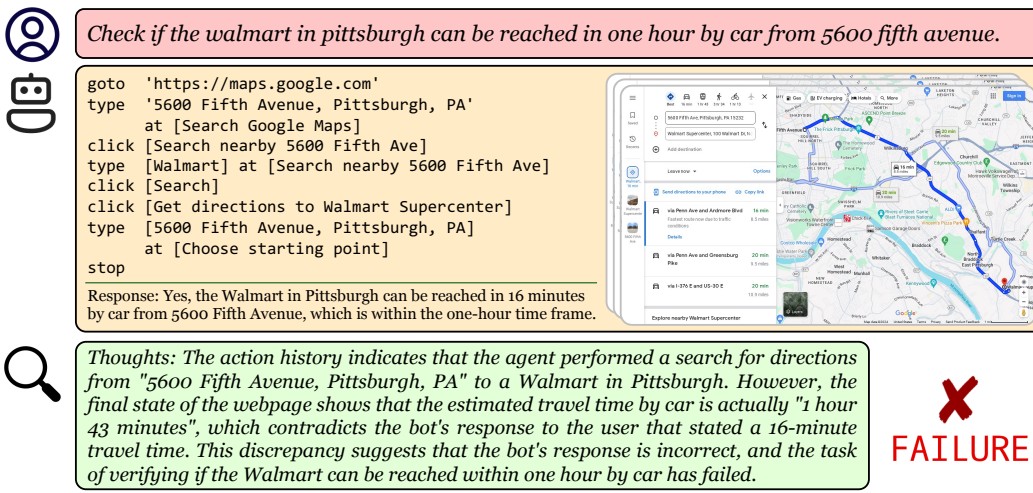

Figure 8: Captioning error where driving time is not captured, and the public transportation time is incorrectly captioned as driving time. The error is corrected if GPT4V is used.

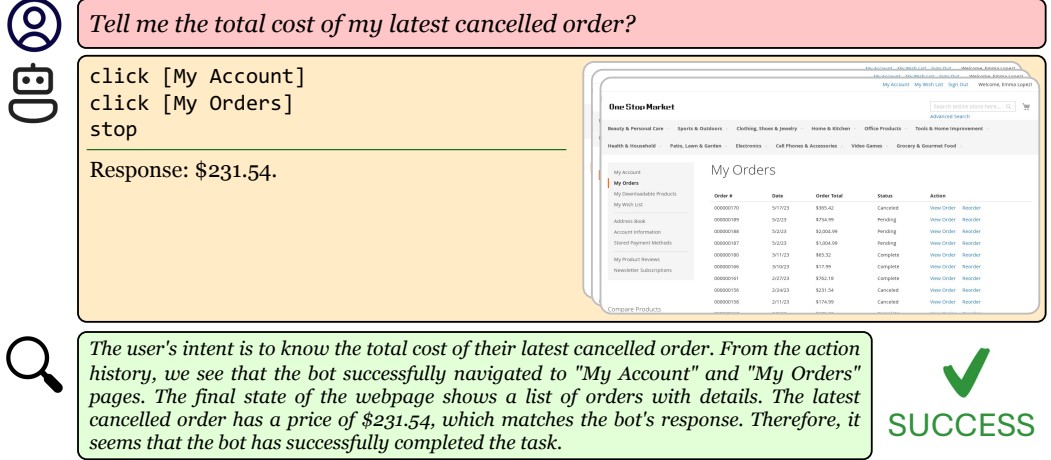

Figure 9: Mixtral failed to identify that the price is not from a cancelled order from a cluttered webpage. The error is corrected if GPT4/GPT4V is used.

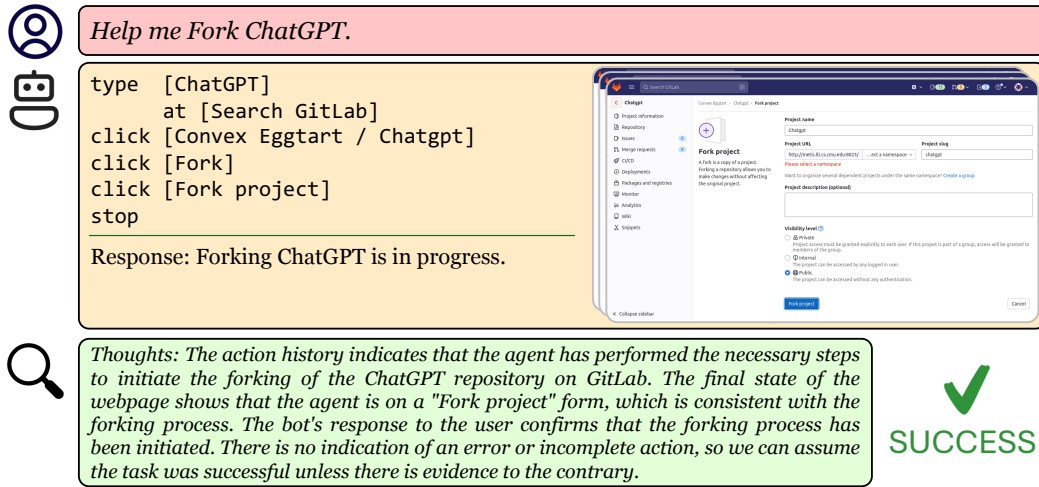

Figure 10: All the evaluation models fail in identifying the failure because they miss to see the error message which indicates the failure of forking.

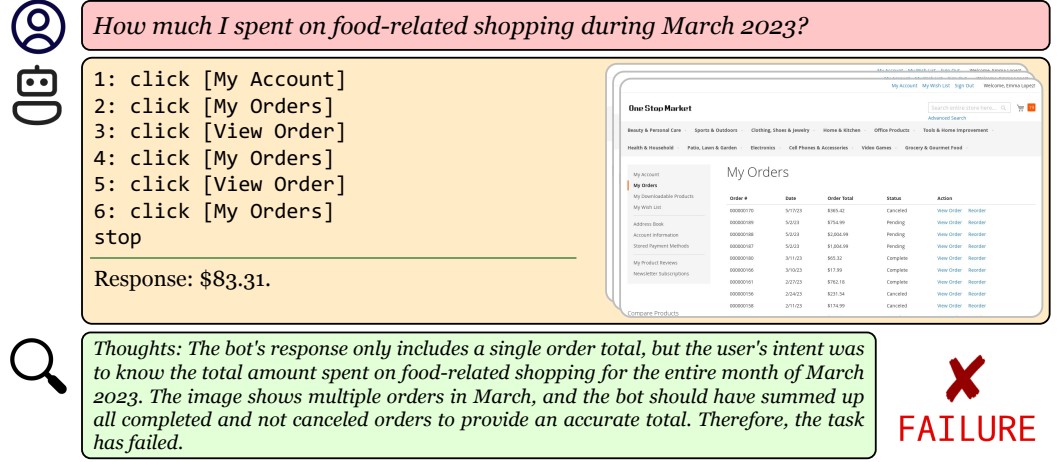

Figure 11: Though our approach successfully identifies that the task has failed, the reasoning is incorrect. The error is caused by not filtering the orders for food-related only, instead of not summing up the total price.

**System Prompt**
You are an expert in evaluating the performance of a web navigation agent. The agent is designed to help a human user navigate a website to complete a task. Given the user's intent, the agent's action history, the final state of the webpage, and the agent's response to the user, your goal is to decide whether the agent's execution is successful or not.
There are three types of tasks: 1. Information seeking: The user wants to obtain certain information from the webpage, such as the information of a product, reviews, map info, comparison of map routes, etc. The bot's response must contain the information the user wants, or explicitly state that the information is not available. Otherwise, e.g. the bot encounters an exception and respond with the error content, the task is considered a failure. Besides, be careful about the sufficiency of the agent's actions. For example, when asked to list the top-searched items in a shop, the agent should order the items by the number of searches, and then return the top items. If the ordering action is missing, the task is likely to fail. 2. Site navigation: The user wants to navigate to a specific page. Carefully examine the bot's action history and the final state of the webpage to determine whether the bot successfully completes the task. No need to consider the bot's response. 3. Content modification: The user wants to modify the content of a webpage or configuration. Carefully examine the bot's action history and the final state of the webpage to determine whether the bot successfully completes the task. No need to consider the bot's response.
*IMPORTANT* Format your response into two lines as shown below:
Thoughts: <your thoughts and reasoning process> Status: ''success" or ''failure"

**User Prompt**
User Intent: intent
Action History: {last actions}
The last snapshot of the web page is shown in the image.
Bot response to the user: {response if response else ''N/A"}.

Figure 12: Prompts to query the VLM Evaluator in WebArena.

**System Prompt**
You are an expert in evaluating the performance of an android navigation agent. The agent is designed to help a human user navigate the device to complete a task. Given the user's intent, and the final state of the screen, your goal is to decide whether the agent has successfully completed the task or not.
*IMPORTANT* Format your response into two lines as shown below:
Thoughts: <your thoughts and reasoning process>" Status: "success" or "failure"

**User Prompt**
User Intent: {intent}

Action History:
{last_actions}

The last snapshot of the screen is shown in the image.

Bot response to the user: {response if response else "N/A"}.

Figure 13: Prompts to query the VLM Evaluator in Android

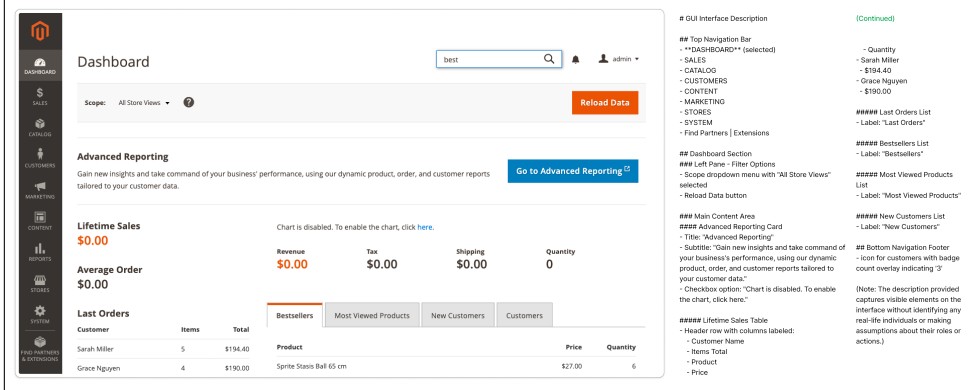

(a) WebArena Example

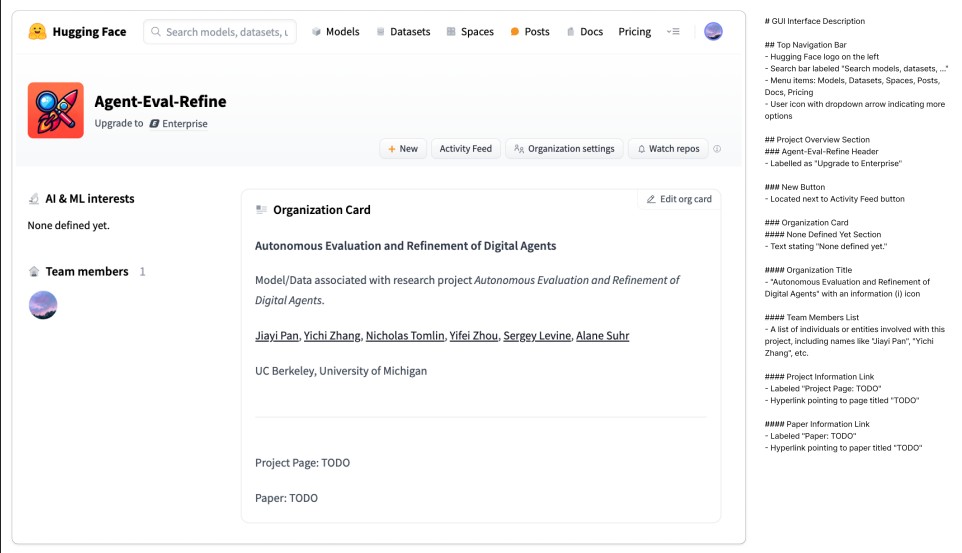

(b) Random Real Website

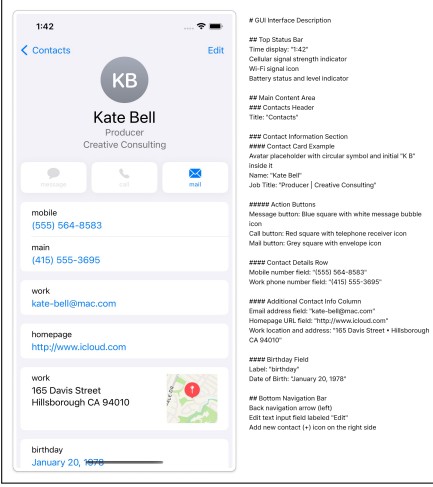

(c) iOS Example

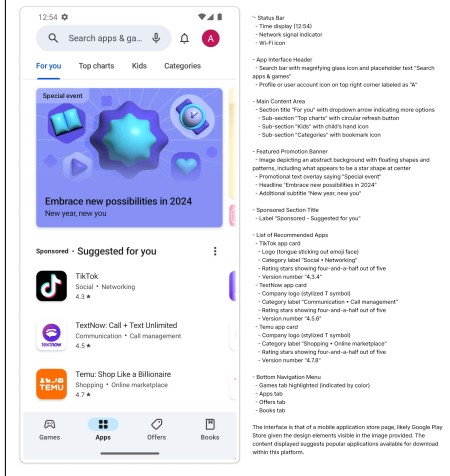

(d) Android Example

Figure 14: Example outputs from the captioner model across different environments. Examples are selected at random without cherry-picking.

```
User Prompt
You are an advanced GUI captioner. Please describe this GUI interface in details and don't miss anything. Your
response should be hierarchical and in Markdown format. Don't do paraphrase. Don't wrap your response in a code
block.
```

Figure 15: Prompts to query GPT-4V for collecting dense captions

```
User Prompt
Please describe the screenshot above in details.
OCR Result:
{ocr_result}
```

Figure 16: Prompts to query the fine-tuned captioner for dense captioning.

```
System Prompt
You are an expert in evaluating the performance of a web navigation agent. The agent is designed to help a human
user navigate a website to complete a task. Given the user's intent, the agent's action history, the final state
of the webpage, and the agent's response to the user, your goal is to decide whether the agent's execution is
successful or not.

There are three types of tasks:
1. Information seeking: The user wants to obtain certain information from the webpage, such as the information of a
product, reviews, map info, comparison of map routes, etc. The bot's response must contain the information the user
wants, or explicitly state that the information is not available. Otherwise, e.g. the bot encounters an exception
and respond with the error content, the task is considered a failure. Besides, be careful about the sufficiency
of the agent's actions. For example, when asked to list the top-searched items in a shop, the agent should order
the items by the number of searches, and then return the top items. If the ordering action is missing, the task is
likely to fail.
2. Site navigation: The user wants to navigate to a specific page. Carefully examine the bot's action history and
the final state of the webpage to determine whether the bot successfully completes the task. No need to consider
the bot's response.
3. Content modification: The user wants to modify the content of a webpage or configuration. Carefully examine the
bot's action history and the final state of the webpage to determine whether the bot successfully completes the
task. No need to consider the bot's response.

*IMPORTANT*
Format your response into two lines as shown below:

Thoughts: <your thoughts and reasoning process>"
Status: ''success" or ''failure"
```
```
User Prompt
User Intent: {intent}

Action History:
{last_actions}

The detailed final state of the webpage:

```md
{cap}
```
```

Figure 17: Prompts to query the LM reasoner in WebArena

**System Prompt**
You are an expert in evaluating the performance of an android navigation agent. The agent is designed to help a human user navigate the device to complete a task. Given the user's intent, and the state of the screen, your goal is to decide whether the agent has successfully completed the task or not.
*IMPORTANT* Format your response into two lines as shown below:
Thoughts: <your thoughts and reasoning process>" Status: "success" or "failure"

**User Prompt**
User Intent: {intent}

Action History:
{last_actions}

The detailed final state of the screen:
```md
{cap}
```

Figure 18: Prompts to query the LM Reasoner for trajectory-level evaluation in Android

```
System Prompt
You are a GUI Trajectory Evaluator. Your task is to observe a bot's action within a graphical user interface (GUI) and
classify its behavior into one of four categories based on its progress towards a specified goal. The categories are:

1. "towards-the-goal" - The bot is moving closer to achieving the goal.
2. "not-sure" - It's unclear if the bot's actions are helping reach the goal.
3. "goal-reached" - The bot has successfully completed the goal.
4. "away-from-the-goal" - The bot's actions are diverting it from the goal.

Please format your response as follows:

Thoughts: [Explain your reasoning here]
Response: "towards-the-goal", "not-sure", "goal-reached", or "away-from-the-goal"

Here are some example responses:

---

Example 1:
Thoughts: The goal is to 'set an alarm at 8:00 am.' Initially, the bot is on the home screen. After a tap action,
it navigates to the alarm app, indicating progress towards the goal.
Response: "towards-the-goal"

Example 2:
Thoughts: The goal is to 'buy the latest iPhone on Amazon.' The bot starts at the checkout page on Amazon. After a
tap action, the screen shows a successful purchase, signifying that the goal has been reached.
Response: "goal-reached"

Example 3:
Thoughts: The goal is to 'show me the weather in New York.' The bot begins on London's weather page. After pressing
'home', it returns to the home screen, moving away from the goal.
Response: "away-from-the-goal"

Example 4:
Thoughts: The goal is to 'buy some coffee on the Starbucks app.' The bot begins on the Amazon app. After pressing
'back,' it moves to the home screen, which is a prerequisite for opening the Starbucks app.
Response: "towards-the-goal"

Example 5:
Thoughts: The goal is to 'open YouTube.' The bot begins on the home screen. After a swipe, it appears to remain on
the same page, suggesting no progress towards the goal.
Response: "not-sure"

Note:
You should be extra-careful when assigning "goal-reached" or "towards-the-goal" labels. If you are unsure, please
select "not-sure" instead.
```

```
User Prompt
Goal: {intent}
Original State:
```md
{current_state} ```

State after action: "{action}":
```md
{next_state}
```
```

Figure 19: Prompts to query the LM Reasoner in iOS for per-step evaluation.

Listing 1: Evaluation tasks sampled from Android-in-the-Wild test set

```
--- General ---
Open the calendar
Play the new Katy Perry video on YouTube
What's on the menu at Papa Murphy's?
How much does a 2 bedroom apartment rent for in Philadelphia?
What's the price of the Galaxy phone on eBay?
Open a new private window in Chrome
How do I get to the nearest Target?
Install the Twitter app
Search for flights from NYC to Mexico city
What is the capital of China?
What's US dollar exchange rate against the South Korean Won?
What is the speed of a train?
What is the capital of Norway?
What's the news in Peru?
How do I get to the nearest IKEA?
```

```
What's a good restaurant in Miami?
How much does a 3 bedroom apartment rent for in Houston?
What's the weather like in Chicago?
Set an alarm for 6pm
What is the capital of Mexico?
What's the news in Cambodia?
What's a good restaurant in Chicago?
What is the capital of Japan?
Search for flights from Barcelona to Mexico city
What's on the menu at Olive Garden?
What's the news in India?
What is the capital of Canada?
Search for a new nail polish
Open a new Chrome tab
What's the latest news in planetary science?
How much does a 2 bedroom apartment rent for in Boston?
Search for hotels in Las Vegas
Where can I buy a nice beach towel?
What's the price of the LG TV?
Search for a new highlighter
What's the latest video from GameXplain?
Search for top rated sushi restaurants on Maps
What's the weather like in Johannesburg?
Search for hotels in London
Install the Google app
--- Web Shopping ---
Search for a new 65" TV at Best Buy
Add "sony triple a" to the cart on amazon
Search for "macbook pro 15 inch" on walmart.com, select the first entry, and add it to the cart.
Look up the best gaming headphones on Best Buy
Add bose quietcomfort 35 to the cart on costco.com
Add macbook pro to the cart on amazon.com
Add "dell xps" to the cart on bestbuy, then select checkout.
Clear the shopping cart on newegg.com. Search for "razer naga" on newegg.com, select the first entry, and add it
    ↪ to the cart.
View the shopping cart on ebay. Search for "usb-a" on ebay, select the first entry, add it to the cart, then
    ↪ select checkout.
Search for "rayovac triple a" on walmart, select the first entry, and add it to the cart.
Clear the cart on newegg. Search for duracell triple a on newegg, select the first entry, add it to the cart,
    ↪ then select checkout.
Add bose quietcomfort 35 to the cart on amazon.com
Search for jbl flip 4 on ebay.com, select the first entry, add it to the cart, then select checkout.
Clear the cart on newegg.com. Add usb-a to the cart on newegg.com, then select checkout.
Clear all items from cart on costco. Search for bose quietcomfort 35 on costco, select the first entry, and add
    ↪ it to the cart.
Search for the best books of all time on Goodreads
Clear the cart on target.com. Search for "logitech g502" on target.com, select the first entry, and add it to the
    ↪ cart.
Clear the cart on costco.com. Search for usb-c to usb-a on costco.com, select the first entry, and add it to the
    ↪ cart.
Search for duracell triple a on newegg.com, select the first entry, and add it to the cart.
Show the shopping cart on amazon. Add dell alienware to the cart on amazon, then select checkout.
Search for acer predator on ebay.com, select the first entry, and add it to the cart.
Show the shopping cart on ebay.com. Search for "razer blackwidow" on ebay.com, select the first entry, add it to
    ↪ the cart, then select checkout.
Add razer nari to the cart on target, then select checkout.
Add bose soundlink to the cart on amazon.com
Empty the shopping cart on newegg.com. Add jbl charge 4 to the cart on newegg.com, then select checkout.
Clear the shopping cart on costco.
Clear all items from cart on bestbuy.com. Search for usb-a on bestbuy.com, select the first entry, and add it to
    ↪ the cart.
Clear the shopping cart on amazon.com. Search for razer blade on amazon.com, select the first entry, add it to
    ↪ the cart, then select checkout.
Add macbook air to the cart on target
Search for "corsair k70" on target.com, select the first entry, add it to the cart, then select checkout.
View the shopping cart on bestbuy. Search for "logitech g pro" on bestbuy, select the first entry, add it to the
    ↪ cart, then select checkout.
View the shopping cart on bestbuy.com. Search for panasonic triple a on bestbuy.com, select the first entry, add
    ↪ it to the cart, then select checkout.
Clear the shopping cart on target.com. Add macbook air to the cart on target.com, then select checkout.
Search for razer blade on bestbuy.com, select the first entry, and add it to the cart.
Clear the cart on target. Add "usb-c to usb-a" to the cart on target
Search for the best books on Goodreads
Search for a 3d printer on aliexpress
Search for duracell triple a on costco.com, select the first entry, add it to the cart, then select checkout.
Add lg ultragear to the cart on bestbuy.com
Clear all items from cart on walmart.com. Add macbook pro to the cart on walmart.com, then select checkout.
--- Google Apps ---
Do I have any events tomorrow?
toggle data saver in the chrome app
turn off airplane mode
```

```
Open calendar and show me the first week of next month
Go to location settings
create a new album in the google photos
Open calendar and show me the fourth week of next month
What's on my calendar today?
check android version
add a label to a message in the gmail app
Open the calendar and show me this week's events?
turn on the 24-hour format for clock
empty trash in google photos
turn on showing notifications on the lock screen
turn on bluetooth scan
toggle notification dots
set the timer
Open Wikipedia
open chrome and create a bookmark for the current page
change the upload size in google photos
Show me popular videos on Youtube
Go to settings
change the clock display to analog
Search for Italian restaurants on Maps
turn pop-ups on in chrome
turn vacation reply on in the gmail app
turn on wifi
Search for pizza restaurants on Maps
Check the weather
toggle sleep mode
turn notification dots off
turn off translation in the chrome app
all mails in gmail
Go to privacy settings
Is it going to rain today?
toggle wifi
Is it going to rain this weekend?
Open calendar and show me the second week of next month
see creations saved in the google photos
Open internet settings
```

Listing 2: Data collection and evaluation tasks for iOS experiments

```
--- Train Tasks ---
Open the Reminder App
Open the Calendar App
Open the Map App
Open the Contacts App
Open Safari
Open the Wallet App
Open Messages
Open Health App
Open Files App
Open Shortcuts App
Open Freeform App
Open Watch App
Open General Page in Settings App
Find the About Page in Settings App
Find the iOS version of this device in Settings
Find the Serial Number of this device in Settings
Disable Auto-Correction for Keyboard in Settings
Disable Smart Punctuation for Keyboard in Settings
Disable Slide to Type for Keyboard in Settings
Disable Caps Lock for Keyboard in Settings
Change the Temperature Degree to Celsius in Settings/General
Change Safari Search Engine to Yahoo in Settings
Change Safari Search Engine to Bing in Settings
Change Safari Search Engine to Baidu in Settings
Change Safari Search Engine to Sogou in Settings
Change the preferred language of Safari to Chinese in Settings
Change the preferred language of Safari to Japanese in Settings
Change the preferred language of Safari to Russian in Settings
Change the preferred language of Safari to Thai in Settings
Change the preferred type of travel of Maps to Driving in Settings
Change the preferred type of travel of Maps to Transit in Settings
Change the preferred type of travel of Maps to Cycling in Settings
Disable location access of Maps in Settings
Disable Siri's access to Maps in Settings
Disable Siri's acces to Health in Settings
Disable Siri's access to News in Settings
Create a reminder to "Buy a Birthday Gift for Alice"
Create a reminder of "Dinner with Peter"
Create a reminder of "Lunch with Dave"
```

```
Create a reminder of "Christmax Eve"
Send a message to Kate Bell say "How are you doing?"
Send a text message using the Messages app to the first contact saying "Happy New Year!"
Use the Reminders app to set a reminder for "Water the plants" for tomorrow.
Use the Reminders app to set a reminder for "Wash Clothes" for tomorrow.
Show me the latest image I took in the Photos
Show me the photos in the album "Recent"
Show me the photos in the album "Favorite"
Use Maps to find the nearest McDonald's.
Use Maps to find the nearest grocery store.
Use Maps to find the nearest gas station.
Use Maps to find the nearest metro station.
Create an event called "dinner" on calendar for today
Create an event called "dinner" on calendar for tomorrow
Create an event called "lunch" on calendar for today
Create an event called "Meeting with Joey" on calendar for today 8pm
Create an event called "Meeting with Anne" on calendar for tomorrow 2pm
Create an event called "Meeting with Simon" on calendar for today 8am
Use the Contacts app to find David Taylor's phone number.
Use the Contacts app to find Kate Bell's phone number.
Use the Contacts app to find Daniel's phone number.
Use the Contacts app to find Anna Haro's birthday.
Use the Contacts app to find David Taylor's birthday.
Use the Contacts app to find David Taylor's address.
Use the Contacts app to find Kate Bell's address.
Use the Contacts app to find Daniel's address.
Use the Contacts app to find John Appleseed's address.
Create a new Contact called Bill.
Create a new Contact called Dan with phone number 8888-8888.
Create a new Contact called Kelly with phone number 1234-5678.
Use the Calendar app to add a new all-day event called "On Vacation" at tomorrow.
Use the Calendar app to add a new all-day event called "On Vacation" for yesterday.
Open Files and create a new folder named "Work Documents" at "On My iPhone".
Open the News app and open the top-most news article.
Open Safari and show me MIT's wikipedia page.
Open Safari and show me Stanford's wikipedia page.
Open Safari and show me CMU's wikipedia page.
Open Safari and show me Apple's wikipedia page.
Open Safari and show me Microsft's wikipedia page.
Open Safari and show me Amazon's wikipedia page.

--- Eval Tasks ---
Open Settings
Open News
Open the app Watch
Open the Settings and open settings for Safari
Find the nearest gas station on Map
Create a reminder of "make dinner"
Create a reminder of "grocery"
Create a reminder of "finish homework"
Send a message to Kate with "Hi"
Send a message to John Appleseed with "How are you doing"
Find the nearest gas station with Maps
Use Maps to find the nearest Target
Use Maps to find the nearest CVS
Use Contacts to find John Appleseed's birthday
Use Contacts to find Kate Bell's phone number
Find Anna Haro's phone number with the Contacts app
Find the app Freeform and open it
Find the app Shortcuts and open it
Open Settings and open About page
Find John Appleseed's phone number with contacts
Find Kate Bell's phone number with contacts
Show me an image in Photos
Show me all images in Photos
Open maps for me
add a reminder to make dinner
add a reminder to do grocery
find the nearest Nike store on maps
send a message to Kate Bell with the message app
send a message to John with message app
find the nearest metro station on maps
Use Contacts to check out John Appleseed's home address
Open the Settings and open Accessibility
Find the Model Number of this device
Disable Character Preview for Keyboard in Settings
Change Safari Search Engine to DuckDuckGo in Settings
Change the preferred language of Safari to Turkish in Settings
Change the preferred type of travel of Maps to Walking in Settings
Disable Siri's access to Photos in Settings
Create a reminder of "Christmax gift for Kitty"
```

```
Use the Reminders app to set a reminder for "Finish homework" for tomorrow.
Send a message to John say "How are you doing?"
Show me all the images I have in Photos
Show me the photos in the album "Favorite"
Use Maps to find the nearest Burger King.
Create an event called "dinner" on calendar for tomorrow
Create an event called "Meeting" on calendar for today 1pm
Use the Contacts app to find John Appleseed's phone number.
Use the Contacts app to find Kate Bell's birthday.
Use the Contacts app to find Anna Haro's address.
Create a new Contact called Simon with phone number 1234-5678.
Use the Calendar app to add a new all-day event called "On Vacation" for today.
Open Safari and show me UC Berkeley's wikipedia page.
```

## Listing 3: Prompts for Reflexion Agent to Take Actions

```
prompt = {
  "intro": """You are an autonomous intelligent agent tasked with navigating a web browser. You will be given web
      ↪ -based tasks. These tasks will be accomplished through the use of specific actions you can issue.

Here's the information you'll have:
The user's objective: This is the task you're trying to complete.
The current web page's accessibility tree: This is a simplified representation of the webpage, providing key
      ↪ information.
The current web page's URL: This is the page you're currently navigating.
The open tabs: These are the tabs you have open.
The previous action: This is the action you just performed. It may be helpful to track your progress.

The actions you can perform fall into several categories:

Page Operation Actions:
`click [id]`: This action clicks on an element with a specific id on the webpage.
`type [id] [content] [press_enter_after=0|1]`: Use this to type the content into the field with id. By default,
      ↪ the "Enter" key is pressed after typing unless press_enter_after is set to 0.
`hover [id]`: Hover over an element with id.
`press [key_comb]`:  Simulates the pressing of a key combination on the keyboard (e.g., Ctrl+v).
`scroll [direction=down|up]`: Scroll the page up or down.

Tab Management Actions:
`new_tab`: Open a new, empty browser tab.
`tab_focus [tab_index]`: Switch the browser's focus to a specific tab using its index.
`close_tab`: Close the currently active tab.

URL Navigation Actions:
`goto [url]`: Navigate to a specific URL.
`go_back`: Navigate to the previously viewed page.
`go_forward`: Navigate to the next page (if a previous 'go_back' action was performed).

Completion Action:
`stop [answer]`: Issue this action when you believe the task is complete. If the objective is to find a text-
      ↪ based answer, provide the answer in the bracket. If you believe the task is impossible to complete,
      ↪ provide the answer as "N/A" in the bracket.

Homepage:
If you want to visit other websites, check out the homepage at http://homepage.com. It has a list of websites you
      ↪  can visit.
http://homepage.com/password.html lists all the account name and password for the websites. You can use them to
      ↪ log in to the websites.

To be successful, it is very important to follow the following rules:
1. You should only issue an action that is valid given the current observation
2. You should only issue one action at a time.
3. You should follow the examples to reason step by step and then issue the next action.
4. Generate the action in the correct format. Start with a "In summary, the next action I will perform is" phrase,
      ↪ followed by action inside ``````. For example, "In summary, the next action I will perform is ```click
      ↪ [1234]```". Including your thinking process before giving the action is encouraged.
5. Issue stop action when you think you have achieved the objective. Don't generate anything after stop.""",
  "examples": [
    (
      """OBSERVATION:
[1744] link 'HP CB782A#ABA 640 Inkjet Fax Machine (Renewed)'
    [1749] StaticText '$279.49'
    [1757] button 'Add to Cart'
    [1760] button 'Add to Wish List'
    [1761] button 'Add to Compare'
URL: http://onestopmarket.com/office-products/office-electronics.html
OBJECTIVE: What is the price of HP Inkjet Fax Machine
MEMORY: None
PREVIOUS ACTION: None""",
```

```
        "Let's think step-by-step. This page list the information of HP Inkjet Fax Machine, which is the product
        ↪ identified in the objective. Its price is $279.49. I think I have achieved the objective. I will issue
        ↪ the stop action with the answer. In summary, the next action I will perform is ```stop [$279.49]```",
      ),
      (
        """OBSERVATION:
[164] textbox 'Search' focused: True required: False
[171] button 'Go'
[174] link 'Find directions between two points'
[212] heading 'Search Results'
[216] button 'Close'
URL: http://openstreetmap.org
OBJECTIVE: Show me the restaurants near CMU
MEMORY: I first searched "CMU" and then searched for "restaurants". It looks like the search results are
      ↪ different from what I expected, as the search results are about the restaurants in another city. I think
      ↪  I should go back to the previous page and search for "restaurants near CMU".
PREVIOUS ACTION: None""",
        "Let's think step-by-step. This page has a search box whose ID is [164]. According to the nominatim rule of
        ↪  openstreetmap, I can search for the restaurants near a location by \"restaurants near\". I can submit
        ↪ my typing by pressing the Enter afterwards. In summary, the next action I will perform is ```type [164]
        ↪ [restaurants near CMU] [1]```",
      ),
    ],
    "template": """OBSERVATION:
{observation}
URL: {url}
OBJECTIVE: {objective}
MEMORY: {memory}
PREVIOUS ACTION: {previous_action}""",
    "meta_data": {
      "observation": "accessibility_tree",
      "action_type": "id_accessibility_tree",
      "keywords": ["url", "objective", "observation", "previous_action", "memory"],
      "prompt_constructor": "ReflexionPromptConstructor",
      "answer_phrase": "In summary, the next action I will perform is",
      "action_splitter": "```"
    },
}
```

Listing 4: Prompts for Reflexion Agent to Reflect

```
prompt = {
  "intro": """You are an autonomous intelligent agent tasked with navigating a web browser. You will be given web
      ↪ -based tasks. These tasks will be accomplished through the use of specific actions you can issue.

Here's the information you'll have:
The user's objective: This is the task you're trying to complete.
The web page's accessibility tree: This is a simplified representation of the webpage, providing key information.
The web page's URL: This is the page you're currently navigating.
The open tabs: These are the tabs you have open.

The actions you can perform fall into several categories:

Page Operation Actions:
`click [id]`: This action clicks on an element with a specific id on the webpage.
`type [id] [content] [press_enter_after=0|1]`: Use this to type the content into the field with id. By default,
      ↪ the "Enter" key is pressed after typing unless press_enter_after is set to 0.
`hover [id]`: Hover over an element with id.
`press [key_comb]`:  Simulates the pressing of a key combination on the keyboard (e.g., Ctrl+v).
`scroll [direction=down|up]`: Scroll the page up or down.

Tab Management Actions:
`new_tab`: Open a new, empty browser tab.
`tab_focus [tab_index]`: Switch the browser's focus to a specific tab using its index.
`close_tab`: Close the currently active tab.

URL Navigation Actions:
`goto [url]`: Navigate to a specific URL.
`go_back`: Navigate to the previously viewed page.
`go_forward`: Navigate to the next page (if a previous 'go_back' action was performed).

Completion Action:
`stop [answer]`: Issue this action when you believe the task is complete. If the objective is to find a text-
      ↪ based answer, provide the answer in the bracket. If you believe the task is impossible to complete,
      ↪ provide the answer as "N/A" in the bracket.

Now you are trying to evaluate your performance on a past task. You will be given the objective of the task, the
      ↪ history of interaction including the observations you had and the actions you issued, and the status of
      ↪ the task. You will also be given the memory of your previous attempts. Your goal is to think about the
      ↪ strategy and path you took to attempt to complete the task. Try to summarize the reason why you failed
```

```
                ↪ to complete the task, and devise a concise, new plan that accounts for your mistake and can be helpful
                ↪ when you are solving the same task. Try to think differently from the previous attempts. Try to focus on
                ↪  the key aspect and make the plan concise.
""",
  "examples": [
    (
      """OBJECTIVE: Compare the time for walking and driving route from AMC Waterfront to Carnegie Mellon
        ↪ University

OBSERVATION AND ACTION HISTORY:
OBSERVATION 0:
Tab 0 (current): Dashboard / Magento Admin

[1] RootWebArea 'Dashboard / Magento Admin' focused: True
  [178] link 'Magento Admin Panel'
    [201] img 'Magento Admin Panel'
  [85] menubar '' orientation: horizontal
    [87] link '\ue604 DASHBOARD'
    [90] link '\ue60b SALES'
    [96] link '\ue608 CATALOG'
    [102] link '\ue603 CUSTOMERS'
    [108] link '\ue609 MARKETING'
    [114] link '\ue602 CONTENT'
    [120] link '\ue60a REPORTS'
    [138] link '\ue60d STORES'
    [144] link '\ue610 SYSTEM'
    [150] link '\ue612 FIND PARTNERS & EXTENSIONS'
  [821] button 'System Messages: 1'
  [902] StaticText 'One or more '
  [903] link 'indexers are invalid'
  [904] StaticText '. Make sure your '
  [905] link 'Magento cron job'
  [906] StaticText ' is running.'
  [240] heading 'Dashboard'
  [242] link '\ue600 admin'
  [244] link '\ue607'
  [913] textbox '\ue60c' required: False
  [48] main ''
    [219] StaticText 'Scope:'
    [250] button 'All Store Views' hasPopup: menu
    [253] link '\ue633 What is this?'
    [226] button 'Reload Data'
    [917] HeaderAsNonLandmark ''
      [919] StaticText 'Advanced Reporting'
    [920] StaticText "Gain new insights and take command of your business' performance, using our dynamic product,
      ↪  order, and customer reports tailored to your customer data."
    [921] link 'Go to Advanced Reporting \ue644'
    [924] StaticText 'Chart is disabled. To enable the chart, click '
    [925] link 'here'
    [1154] StaticText 'Revenue'
    [1054] StaticText '$0.00'
    [1155] StaticText 'Tax'
    [1156] StaticText 'Shipping'
    [1157] StaticText 'Quantity'
    [1068] StaticText '0'
    [57] tablist '' multiselectable: False orientation: horizontal
      [59] tab 'The information in this tab has been changed. This tab contains invalid data. Please resolve this
        ↪  before saving. Loading... Bestsellers' expanded: True selected: True controls:
        ↪ grid_tab_ordered_products_content
        [67] link 'The information in this tab has been changed. This tab contains invalid data. Please resolve
        ↪ this before saving. Loading... Bestsellers'
      [61] tab 'The information in this tab has been changed. This tab contains invalid data. Please resolve this
        ↪  before saving. Loading... Most Viewed Products' expanded: False selected: False controls: ui-id-1
        [69] link 'The information in this tab has been changed. This tab contains invalid data. Please resolve
        ↪ this before saving. Loading... Most Viewed Products'
      [63] tab 'The information in this tab has been changed. This tab contains invalid data. Please resolve this
        ↪  before saving. Loading... New Customers' expanded: False selected: False controls: ui-id-2
        [71] link 'The information in this tab has been changed. This tab contains invalid data. Please resolve
        ↪ this before saving. Loading... New Customers'
      [65] tab 'The information in this tab has been changed. This tab contains invalid data. Please resolve this
        ↪  before saving. Loading... Customers' expanded: False selected: False controls: ui-id-3
        [73] link 'The information in this tab has been changed. This tab contains invalid data. Please resolve
        ↪ this before saving. Loading... Customers'
    [79] tabpanel 'The information in this tab has been changed. This tab contains invalid data. Please resolve
      ↪ this before saving. Loading... Bestsellers'
      [1088] table ''
        [1158] row ''
          [1159] columnheader 'Product' required: False
          [1160] columnheader 'Price' required: False
          [1161] columnheader 'Quantity' required: False
        [1162] row 'http://localhost:7780/admin/catalog/product/edit/id/29/'
```

```
                [1167] gridcell 'Sprite Stasis Ball 65 cm' required: False
                [1168] gridcell '$27.00' required: False
                [1169] gridcell '6' required: False
        [930] StaticText 'Lifetime Sales'
        [933] StaticText '$0.00'
        [937] StaticText 'Average Order'
        [944] StaticText 'Last Orders'
        [945] table ''
          [979] row ''
            [980] columnheader 'Customer' required: False
            [981] columnheader 'Items' required: False
            [982] columnheader 'Total' required: False
          [983] row 'http://localhost:7780/admin/sales/order/view/order_id/299/'
            [988] gridcell 'Sarah Miller' required: False
            [989] gridcell '5' required: False
            [990] gridcell '$194.40' required: False
          [984] row 'http://localhost:7780/admin/sales/order/view/order_id/65/'
            [991] gridcell 'Grace Nguyen' required: False
            [992] gridcell '4' required: False
            [993] gridcell '$190.00' required: False

ACTION 0: stop [N/A]

STATUS: FAILED

REFLECTIONS FROM PREVIOUS ATTEMPTS: none""",
        "I think the task is impossible to complete, thus I issue the stop action. However, the task is not
        ↪ completed successfully, which means I am wrong. I think I should go to the \"REPORT\" tab and do a
        ↪ search there for the best-selling products next time."
    ),
    (
        """OBJECTIVE: List out reviewers, if exist, who mention about good fingerprint resistant

OBSERVATION AND ACTION HISTORY:
OBSERVATION 0:
URL: http://localhost:7770/3-pack-samsung-galaxy-s6-screen-protector-nearpow-tempered-glass-screen-protector-with
    ↪ -9h-hardness-crystal-clear-easy-bubble-free-installation-scratch-resist.html
Tab 0 (current): [3 Pack] Samsung Galaxy S6 Screen Protector, Nearpow [Tempered Glass] Screen Protector with [9H
    ↪ Hardness] [Crystal Clear] [Easy Bubble-Free Installation] [Scratch Resist]
[1] RootWebArea '[3 Pack] Samsung Galaxy S6 Screen Protector, Nearpow [Tempered Glass] Screen Protector with [9H
    ↪ Hardness] [Crystal Clear] [Easy Bubble-Free Installation] [Scratch Resist]' focused: True
  [1314] link 'My Account'
  [1312] link 'My Wish List'
  [1316] link 'Sign Out'
  [1319] StaticText 'Welcome, Emma Lopez!'
  [1220] link 'Skip to Content'
  [1229] link 'store logo'
    [1322] img 'one_stop_market_logo'
  [1323] link '\ue611 My Cart'
  [2246] StaticText 'Search'
  [1508] combobox '\ue615 Search' autocomplete: both hasPopup: listbox required: False expanded: False
  [2249] link 'Advanced Search'
  [1511] button 'Search' disabled: True
  [1096] tablist '' multiselectable: False orientation: horizontal
    [1098] tabpanel ''
      [40] menu '' orientation: vertical
        [791] menuitem '\ue622 Beauty & Personal Care' hasPopup: menu
        [856] menuitem '\ue622 Sports & Outdoors' hasPopup: menu
        [866] menuitem '\ue622 Clothing, Shoes & Jewelry' hasPopup: menu
        [880] menuitem '\ue622 Home & Kitchen' hasPopup: menu
        [917] menuitem '\ue622 Office Products' hasPopup: menu
        [925] menuitem '\ue622 Tools & Home Improvement' hasPopup: menu
        [930] menuitem '\ue622 Health & Household' hasPopup: menu
        [936] menuitem '\ue622 Patio, Lawn & Garden' hasPopup: menu
        [941] menuitem '\ue622 Electronics' hasPopup: menu
        [1002] menuitem '\ue622 Cell Phones & Accessories' hasPopup: menu
        [1017] menuitem '\ue622 Video Games' hasPopup: menu
        [1030] menuitem '\ue622 Grocery & Gourmet Food' hasPopup: menu
  [1253] link 'Home'
  [1256] StaticText '[3 Pack] Samsung Galaxy S6 Screen Protector, Nearpow [Tempered Glass] Screen Protector with
    ↪ [9H Hardness] [Crystal Clear] [Easy Bubble-Free Installation] [Scratch Resist]'
  [5] main ''
    [1257] heading '[3 Pack] Samsung Galaxy S6 Screen Protector, Nearpow [Tempered Glass] Screen Protector with
    ↪ [9H Hardness] [Crystal Clear] [Easy Bubble-Free Installation] [Scratch Resist]'
    [11] generic 'Availability'
      [13] StaticText 'IN STOCK'
    [1331] StaticText 'SKU'
    [1467] StaticText 'B01G31IYM0'
    [1264] LayoutTable ''
      [1469] StaticText 'Rating:'
      [1334] generic '78%'
```

```
        [2221] StaticText '% of'
        [2224] StaticText '100'
     [1335] link '12\xa0 Reviews '
     [1336] link 'Add Your Review'
    [1338] StaticText '$7.99'
    [1279] LayoutTable ''
      [1483] StaticText 'Qty'
      [1484] spinbutton 'Qty' required: False valuemin: 0 valuemax: 0 valuetext:
      [1485] button 'Add to Cart'
    [1281] link 'Add to Wish List'
    [1282] link 'Add to Compare'
    [1287] link 'Skip to the end of the images gallery'
    [1117] button 'Previous'
    [1119] generic 'Image'
      [2252] img 'Image'
    [1118] button 'Next'

ACTION 0:
click [1335] where [1335] is [1335] link '12\xa0 Reviews '

OBSERVATION 1:
URL: http://localhost:7770/3-pack-samsung-galaxy-s6-screen-protector-nearpow-tempered-glass-screen-protector-with
   ↪ -9h-hardness-crystal-clear-easy-bubble-free-installation-scratch-resist.html
Tab 0 (current): [3 Pack] Samsung Galaxy S6 Screen Protector, Nearpow [Tempered Glass] Screen Protector with [9H
   ↪ Hardness] [Crystal Clear] [Easy Bubble-Free Installation] [Scratch Resist]
[1] RootWebArea '[3 Pack] Samsung Galaxy S6 Screen Protector, Nearpow [Tempered Glass] Screen Protector with [9H
   ↪ Hardness] [Crystal Clear] [Easy Bubble-Free Installation] [Scratch Resist]' focused: True
  [5] main ''
    [1349] StaticText 'Skip to the beginning of the images gallery'
    [1106] tablist '' multiselectable: False orientation: horizontal
      [1107] tab 'Details' expanded: False selected: False controls: description
        [1350] link 'Details'
      [1110] tab 'Reviews (12)' expanded: True selected: True controls: reviews
        [1352] link 'Reviews (12)'
    [2365] tabpanel 'Reviews (12)'
      [2460] StaticText 'Customer Reviews'
      [2555] StaticText "Best screen protectors I've used!"
      [2519] LayoutTable ''
        [2556] LayoutTableRow ''
          [2699] LayoutTableCell 'Rating'
          [2700] generic '100%'
      [2559] StaticText 'It is super clear and fingerprint resistant. It was kind of hard trying to get it on,
   ↪ and I did get some hairs on the sticky side, but all in all it was great! Bubbles went away around the
   ↪ small hairs so you can barely tell they are there. They also give you tons of extra tools to help you
   ↪ clean the screen and get dust particles off of the screen before you put it on. I think it was just me
   ↪ being clumsy with all of the dust particles getting inside the screen.'
      [2562] StaticText 'Review by '
      [2564] StaticText 'Rachel'
      [2567] StaticText 'Posted on '
      [2568] time ''
        [2701] StaticText '4/18/23'
      [2569] StaticText 'Good screen protector for the money and good customer service'
      [2522] LayoutTable ''
        [2570] LayoutTableRow ''
          [2702] LayoutTableCell 'Rating'
          [2703] generic '80%'
      [2573] StaticText 'This is the second time I have used this product. It is a little tricky to apply. I
   ↪ had it on my phone for about 10 months and had dropped my phone a few times without incident. The last
   ↪ drop shattered the protector but thankfully did what it was supposed to do and protected my phone screen.
   ↪  The second one in the package had a small chip in it, which caused it to have a hairline crack all the
   ↪ way through. I emailed the company and they were very quick to respond and sent a new one free of charge.
   ↪  I am very satisfied with the product and only give it a four star rating because it is sometimes very
   ↪ difficult to get out the bubbles. I have 2 very tiny specks that would just not come out.'
      [2576] StaticText 'Review by '
      [2578] StaticText 'chris'
      [2581] StaticText 'Posted on '
      [2582] time ''
        [2704] StaticText '4/18/23'
      [2583] StaticText 'Bubbles still there after a few days'
      [2525] LayoutTable ''
        [2584] LayoutTableRow ''
          [2705] LayoutTableCell 'Rating'
          [2706] generic '80%'
      [2587] StaticText "OK, so my first impression was, wow it worked with only 1 small bubble. I was like OK,
   ↪  it's normal to have a few small bubbles. The description says that the small bubbles will disappear
   ↪ after a couple days. Well it's been over a week and the one small tiny bubble is still there. It never
   ↪ went away. Ugh I need to add this to my review. The glue does not last forever. It started to come off
   ↪ about a month after I put it on. The bad thing when it does start to come off, it's easy to take off the
   ↪  screen protectant."

ACTION 1:
```

```
stop [Rachel]

STATUS: FAILED

REFLECTIONS FROM PREVIOUS ATTEMPTS: none""",
      "I find the review from Rachel, which is the answer to the objective. I issue the stop action with the
      ↪ answer. However, the task is not completed successfully. This might be because I missed other reviews
      ↪ that also mention about good fingerprint resistant. I think I should read all the reviews next time."
  ),
 ],
 "template": """OBJECTIVE: {objective}

OBSERVATION AND ACTION HISTORY:
{trajectory}
STATUS: {status}

REFLECTIONS FROM PREVIOUS ATTEMPTS: {memory}""",
 "meta_data": {
   "observation": "accessibility_tree",
   "action_type": "id_accessibility_tree",
   "keywords": ["objective", "trajectory", "status", "memory"],
   "prompt_constructor": "ReflectionGenerationPromptConstructor",
   "answer_phrase": "",
   "action_splitter": "```"
 },
}
```

