# OpenReview forum: "Autonomous Evaluation and Refinement of Digital Agents"
_colmweb.org/COLM/2024/Conference — COLM_

### Official Review · Reviewer_tJw6 · 2024-05-07

**Rating:** 7
**Confidence:** 4
**Ethics Flag:** 1

**Summary:**

This paper tackles an important problem of agentic research: how to automate the evaluation of domain-general agents? Current agent benchmarks rely heavily on hard-coded rules for evaluation, which is expensive to construct and hard to scale. This paper proposes a multimodal approach to automate this process. They first train a VLM-based Captioner to generate a textual description of the agent's observation (i.e., the screenshot). Then the LLM-based Reasoner will take this description along with the actions, and the original user
instruction to judge whether the execution is successful or not.

Given such an automatic evaluator, they further show that it can be used to improve the performance of existing agents by providing inference-time guidance for self-refinement.

**Questions To Authors:**

What would you perceive as the biggest challenge in automating agent evaluation?

**Reasons To Accept:**

- This paper is well-motivated and clearly presented.
- The problem being tackled is important and impactful.
- The proposed approach, although quite simple, is effective on two evaluated benchmarks.

**Reasons To Reject:**

- The key to Captioner's success is the collection of 1,272 manually curated caption annotations, whose collection is slow and expensive. Is there any way to automate this process such that it can be scaled up?
- The proposed approach, despite being effective on WebArena and AitW, might not work well in domains where screenshots cannot capture the change of state. For instance, if an agent executes "mv file_a foler_b" in the terminal. The action will be executed completely in the backend and nothing will be shown unless both folders are explicitly open and displayed on the screen.

---

> ### Author Rebuttal · Authors · 2024-05-31
>
> Thank you for your review! We respond to your points below and will incorporate them into our paper.
>
> > key to Captioner's success is…1,272 manually curated caption annotations...slow and expensive
>
> We’d like to correct a possible misunderstanding. **Collecting these annotations is actually cheap and efficient**. The cost for querying GPT-4V for initial annotation is about $40, and the manual coarse filtering process took only around 2 hours.
>
> Secondly, this caption dataset is only used to teach the model to generate captions with appropriate granularity and correct markdown format. Motivated by recent studies on fine-tuning language models ([Zhou. 2023](https://arxiv.org/abs/2305.11206), [Gudibande. 2023](https://arxiv.org/abs/2305.15717)), we suspect Captioner’s performance is bottlenecked by base model’s GUI understanding capability learnt during pre-training, rather than the size of the fine-tuning dataset (i.e., adding more fine-tuning data has diminishing returns). With open VLM’s capabilities continuously improving, we are optimistic that they should be able to perform the captioning task in zero- or few-shot in the future, similar to proprietary models such as GPT-4V.
>
> > screenshots cannot capture the change of state
>
> Digital environments are designed to be user-friendly, displaying crucial information directly on the screen, such as the consequence of an action. For example, `mv file_a folder_b` will return an error message in the terminal if it fails and otherwise nothing. Thus, trajectory (screenshots and actions) should still be adequate for evaluation in well-designed environments. Part of the motivation behind using the agent trajectory as input is to provide models with the same amount of evidence as a human judge.
>
> However, we agree there will be cases where there’s insufficient evidence in trajectory to determine the agent's success. One promising way to address this is to train or incentivize agents to show evidence of success, e.g., by rewarding the agent to `ls .` and `ls folder_b` to show the contents of the two folders after the task has been completed.
>
>
> > biggest challenge in automating agent evaluation
>
> We think the biggest challenge is in evaluating agents actually deployed in real-world systems at scale. Our work moves towards this goal of going beyond manually-crafted environments by providing reproducible and cost-effective evaluations that align closely with oracle judges. We are happy to discuss other challenges as well!

---

### Official Review · Reviewer_zFha · 2024-05-09

**Rating:** 7
**Confidence:** 4
**Ethics Flag:** 1

**Summary:**

This paper designs two lines of automatic evaluation for digital agents: (1) Using VLMs to caption the trajectory and then prompting the LLM to reason if the trajectory is correct. (2) Using an end-to-end VLM to evaluate the trajectory (GPT-4V in the paper). These methods show a high correlation with ground truth evaluation metrics, and the paper demonstrates that the agents can be improved by using reflective prompting or filtered BC.

**Questions To Authors:**

I think the methods in this paper are promising for digital agent development. But the weaknesses on originality and evaluation may limit the contribution of this paper. I would like to raise my assessment for the paper in the followup phase if the authors could well address the issues and make a rounded discussion.

**Reasons To Accept:**

- The paper is well-written and easy to follow.
- The framework is interesting and holds promise for digital agent development. The paper has the potential to provide insights for the community.
- The experiments involving filtered BC are interesting and address the data scarcity issue in some agent learning settings well.
- The paper curates data and tasks in the iOS domain, which may benefit future studies on digital agents.

**Reasons To Reject:**

- Using LLMs to evaluate the trajectory of agent policies and improving the agent seems common in existing literature. For example, [1] also uses LLM evaluators to conduct filtered BC, and [2] uses reflection.
- The evaluation of filtered BC in the iOS domain is not very persuasive because Table 2 only shows the comparison of the number of successful tasks for different methods. It is unclear if solving 14 out of 52 tasks represents a real improvement or just randomness.

[1] Du et al., Vision-Language Models as Success Detectors.

[2] Wang et al., JARVIS-1: Open-world Multi-task Agents with Memory-Augmented Multimodal Language Models.

---

> ### Author Rebuttal · Authors · 2024-05-31
>
> Thank you for your review! We respond to your points below and will incorporate them into our paper.
>
> > evaluation of filtered BC in the iOS domain is not very persuasive
>
> Scaling experiment in real-world tasks is challenging, shown in recent works ([Du. 2023](https://proceedings.mlr.press/v232/du23b.html) and [Zheng. 2024](https://arxiv.org/abs/2401.01614) use 20 and 92 evaluation tasks respectively). The iOS experiment is limited by requiring physical Apple machines and slow inference speeds of CogAgent, taking 4 physical days for the whole experiment, thus we chose an evaluation set of 52 tasks.
>
> However, to address this concern, we apply filtered BC to AitW (unlike iOS, we can use Android emulators on clusters). We use 560 tasks for training and 96 for evaluation. The base policy (AutoUI-base) completes 15 tasks, improving to 18.9 ± 1.0 (averaged over 3 training runs) with self-training. Filtered BC with our step-wise evaluator still significantly improves, succeeding at 26 ± 0.8 tasks.
>
> > seems common in existing literature
>
> Thank you for highlighting these relevant works; we will add a discussion of them into our paper. We clarify that we don't claim novelty in proposing LM-as-judge or in applying these inference and training techniques.
>
> Our core contribution is to provide strong and comprehensive evidence on the effectiveness of domain-general evaluators in the realistic digital agent domain. Current evaluations of digital agents, conducted in hand-crafted simulators or through reference-based metrics, suffer from labor intensity and limited generalizability. We characterize the pros and cons, and explore the opportunities of using autonomous evaluators. Additionally, we apply exploration-driven learning techniques like filtered-BC to train language agents, in contrast to existing methods that rely on either training with expert demonstrations or inference-time algorithms.
>
> We distinguish ourselves to [1, 2] in
> - Digital agents domain is more open-ended and complex than pick-and-place or insert-remove tasks [1] or plans generated by another agent [2], allowing for arbitrary user instructions across broad domains.
> - Unlike [1], we don’t need any human demonstrations or success labels to train our evaluator, which also mitigates performance degradation in out-of-distribution cases.
> - We experiment with using our evaluators to evaluate policies of varying quality, in contrast to just using the evaluators in a training/refinement loop as in [2].

---

> > ### Comment · Reviewer_zFha · 2024-06-03
> >
> > The authors,
> >
> > Thank you for your responses. My concerns are addressed and I would like to raise the score. It would be better if the discussion and the new results for the rebuttal phase can be included in the revised paper, and the empirical experiments can be more organized.
> >
> > Reviewer zFha

---

> > > ### Author Response · Authors · 2024-06-04
> > >
> > > Dear Reviewer zFha,
> > >
> > > Thanks again for your helpful feedback! We will definitely incorporate all the discussions and new results into the paper, and improve the organization of the experiments.

---

### Official Review · Reviewer_saqr · 2024-05-13

**Rating:** 6
**Confidence:** 4
**Ethics Flag:** 1

**Summary:**

This paper empirically studies model's automatic evaluation of trajectories on digital agents, such as agents operating on the web and Android. The authors focus on two methods to construct the evaluator, one is an end-to-end approach that directly uses a pretrained VLM, another is a modular approach that first transcribes the screenshot to captions with a VLM, and then feeds all text to a LM. Both methods demonstrate high accuracies against oracle evaluator and human judge.  The authors then utilize such automatic evaluators to improve the policy agent, through reflexion and filtered behavioral cloning respectively. In both settings, the evaluation is able to improve the tasks.

**Questions To Authors:**

1. The high accuracy in Table 1 is nice, but I am feeling this is because the special properties of the two environments – for example, the final screenshot is typically enough to classify whether it satisfies the goal. I wonder whether the authors think such high accuracies are general, and whether there are environments where the final state is difficult to classify, I would like to see more discussions on this aspect as potential limitations of the paper.
2. Webarena by design is a text-only task originally, I see the authors are using GPT4-V on that, are the authors experimenting with VisualWebArena or am i missing sth?

```
After rebuttal, I bump up my score to 6 given the clarifications and added results to increase the significance of the previous results.
```

**Reasons To Accept:**

1. Empirically studying whether current models are able to evaluate digital agents’ trajectory success answers a previously unknown question and is useful practically as well.
2. The authors successfully utilize the model evaluators to improve the performance on two difficult tasks, with different approaches.
3. This paper studies both end-to-end and modular approaches, across both proprietary and open-source models, which can provide practical guidance for different readers.

**Reasons To Reject:**

As an empirical paper, I expect the experiments to be conducted in a more comprehensive manner with more details, and in general the empirical results are relatively sparse.  Some examples:
(1) while the authors talk about both trajectory-level and step-level evaluators, step-level evaluation is only applied for the iOS agent refinement, and there is no study how accurate such step-level evaluation is. From Table 2, I guess step-level evaluation may not be very accurate given small improvement over self-training;
(2) end-to-end methods only evaluate GPT-4V, while I would like to see the performance of more open VLMs;
(3) In Table 2, filtered BC only finishes 3 more tasks than self-training in a test of 52 tasks. I am not sure how significant this result is.

---

> ### Author Rebuttal · Authors · 2024-05-31
>
> Thank you for your review! We respond to your points below and will incorporate them into our paper.
>
> > step-level evaluation is only applied for the iOS agent refinement
>
> We focus on using step-level feedback for training via filtered BC, not as an evaluation metric. We didn’t apply this learning approach to WebArena due to the lack of a strong open-weight model in this domain.
>
> > how accurate such step-level evaluation is
>
> Comprehensive evaluation of step-wise metrics requires significantly more hand annotation than overall trajectory success. However, human annotators agreed with our step-wise evaluator for 43 of 50 state-action pairs randomly sampled from the iOS experiments.
>
> > how significant [filtered BC] result is
>
> Please see our response to Reviewer zFha.
>
> > end-to-end methods only evaluate GPT-4V
>
> Preliminary results showed no existing open VLM can simultaneously perform strong reasoning and GUI understanding. This motivated our caption-then-reason approach, which decouples the two capabilities into separate models.
>
> However, to address this concern, we evaluated QWen-VL-chat (the base model for our Captioner and strongest open VLM at the time) as an end-to-end evaluator. It significantly underperforms the open-weight Captioner + Mixtral (from 74.4 to 68.0% accuracy in WebArena; 92.9 to 70.2% in AitW). We didn’t proceed with full agent refinement experiments here given its worse performance.
>
> >  where the final state is difficult to classify
>
> Our method supports feeding intermediate states as input to the evaluator, though this wasn’t helpful in preliminary experiments. We are optimistic that as VLMs perform better long-context reasoning, evaluators will also perform well in cases where the final state alone is hard to classify; e.g. when booking a flight and the confirmation page lacks all flight details.
>
> > experimenting with VisualWebArena?
>
> We use WebArena, which natively supports both text or pixel-based observation space.

---

> > ### Comment · Reviewer_saqr · 2024-06-04
> >
> > Thank you for the clarifications! The added results on AitW is nice and significant, I hereby increase my score.

---

### Decision · Program_Chairs · 2024-07-10

**Decision:**

Accept

**Comment:**

This paper shows that domain-general LLMs are sufficiently good reward models to be able to improve web agents. It has certain weaknesses. It is primarily an empirical paper, and the range and scale of the empirical evaluations seems limited compared to other domains (in part because of the difficulty of scaling evaluations in this domain). And the methods studied are similar to or the same as existing methods. Nevertheless, this is a good study and a well-written paper in an area of intense current interest. It makes real progress in showing that domain-general evaluators (LLMs or a good VLM, really only GPT-4V currently) can be quite effective for general tasks in the web agent domain, for either inference-time guidance or behavior cloning, obviating the need for hand-designed evaluation functions. The transfer learning extension to iOS, while smaller in scale than would be ideal, nevertheless seems a very useful contribution to have.